# Accelerating Monte Carlo Tree Search with Probability Tree State Abstraction

**Yangqing Fu**    **Ming Sun**    **Buqing Nie**    **Yue Gao**[*]
MoE Key Lab of Artificial Intelligence, AI Institute, Shanghai Jiao Tong University
{frank79110, mingsun, niebuqing, yuegao}@sjtu.edu.cn

## Abstract

Monte Carlo Tree Search (MCTS) algorithms such as AlphaGo and MuZero have achieved superhuman performance in many challenging tasks. However, the computational complexity of MCTS-based algorithms is influenced by the size of the search space. To address this issue, we propose a novel probability tree state abstraction (*PTSA*) algorithm to improve the search efficiency of MCTS. A general tree state abstraction with path transitivity is defined. In addition, the probability tree state abstraction is proposed for fewer mistakes during the aggregation step. Furthermore, the theoretical guarantees of the transitivity and aggregation error bound are justified. To evaluate the effectiveness of the *PTSA* algorithm, we integrate it with state-of-the-art MCTS-based algorithms, such as Sampled MuZero and Gumbel MuZero. Experimental results on different tasks demonstrate that our method can accelerate the training process of state-of-the-art algorithms with $10\% - 45\%$ search space reduction.

## 1 Introduction

With the advancement of reinforcement learning (RL), AlphaGo is the first algorithm that can defeat human professional players in the game of Go [1]. With no supervised learning of expert moves in AlphaGo Zero [2] to restructured self-play in AlphaZero [3], until the recent MuZero that conducts Monte Carlo Tree Search (MCTS) in hidden state space with a learned dynamics model [4]. MuZero presents a powerful generalization framework that allows the algorithm to learn without a perfect simulator in complex tasks. Recently, EfficientZero [5] has made significant progress in the sample efficiency of MCTS-based algorithms. This development has opened up new possibilities for real-world applications, including robotics and self-driving.

When dealing with complex decision-making problems, increasing the search depth is necessary to achieve more accurate exploration in the decision space, but this also leads to higher time and space complexity [6, 7]. For instance, MuZero trained for 12 hours with 1000 TPUs to learn the Go game, and for a single Atari game, it needs 40 TPUs for 12 hours of training [4]. One approach to reduce the computation is the state abstraction method, which aggregates states based on a certain similarity measure to obtain a near-optimal policy [8, 9, 10]. state abstraction is a crucial technique in reinforcement learning (RL) that enables efficient planning, exploration, and generalization [11].

To reduce the search space of MCTS, previous studies have shown the potential of specific state abstraction techniques [12, 13, 14, 15]. However, finding the minimum abstract state space in these studies is an NP-Hard problem [16]. To our knowledge, this work is the first to define path transitivity in the formulation of tree state abstraction, which enables the discovery of the minimum abstract state space in polynomial time. Additionally, recent MCTS-based algorithms utilize neural networks to

---

[*]Corresponding author.
Code available at `https://github.com/FYQ0919/PTSA-MCTS`

estimate the value or reward of states, which may lead to errors in aggregating states with deterministic state abstraction functions. To address this issue, we proposed a probability tree state abstraction function that aggregates states based on the distribution of child node values, which enhances the robustness of aggregation and ensures transitivity.

This paper proposes the probability tree state abstraction (*PTSA*) algorithm to improve the tree search efficiency of MCTS. The main contributions can be summarized as follows: i) A general tree state abstraction is formulated, and path transitivity is also defined in the formulation. ii) The probability tree state abstraction is proposed for fewer mistakes during the aggregation step. iii) The theoretical guarantees of the transitivity and aggregation error bound are justified. iv) We integrate *PTSA* with state-of-the-art algorithms and achieve comparable performance with $10\% - 45\%$ reduction in tree search space.

## 2 Related Work

### 2.1 MCTS-based Methods

MCTS is a rollout algorithm for solving sequential decision problems [17]. The fundamental idea of MCTS is to search for the most promising actions by randomly sampling the search space, and then expanding the search tree based on those actions [18]. The computational bottlenecks arise from the search loop, especially interacting with the real environment model of each iteration.

Combined with deep neural networks, MCTS-based methods have achieved better performance and efficiency in various complex tasks, such as board games [2], autonomous driving [19], and robot planning [20]. The model-based algorithm *MuZero* [4] predicts the environmental dynamics model for more efficient simulation. Based on *MuZero*, *EfficientZero* [5] is proposed for the training with limited data, which achieves super-human performance on Atari 100K benchmarks. However, both *MuZero* and *EfficientZero* require high computational consumption when dealing with complex action spaces. To address arbitrarily complex action spaces, the sample-based policy iteration framework [21] is proposed. *Sampled MuZero* extends *MuZero* by sub-sampling a small fraction of possible moves and achieves higher sample efficiency with fewer expanded actions and simulations. The experimental results show that planning over the sampled tree provides a near-optimal approximation [21]. To further reduce the number of simulations, *Gumbel MuZero* utilizes the Gumbel-Top-k trick to construct efficient planning [22].

### 2.2 State Abstraction

State abstraction is aimed at reducing the complex state space by aggregating the similar states [11]. The original state space $S$ can be mapped into a smaller abstract state space $S_\phi$ by state abstraction. By grouping similar states together, state abstraction can help to identify patterns and regularities in the environment, which can inform more effective decision-making [12].

There are two main challenges when applying state abstraction to RL problems. The first challenge is to decrease the value loss between $S$ and $S_\phi$. With bounded value loss, the approximate state abstractions allow the agent to learn a near-optimal policy with improved training efficiency [11, 23]. The second challenge is to compute the smallest possible abstract state space, which is proven that the computational complexity is NP-hard [16]. The transitive state abstraction [11] is defined to efficiently compute the smallest possible abstract state space. However, most transitive state abstractions with deterministic predicates have low fault tolerance. To improve the robustness of aggregation, we measure the abstraction probability of the state pairs based on the expected value distributions. In addition, some previous studies have analyzed and discussed some specific state abstractions in tree structure [12, 13, 14, 15], but there is no formal definition and analysis for general tree state abstractions. To our knowledge, *PTSA* is the first method that defines a general formulation of tree state abstractions for deep MCTS-based methods and analyzes the transitivity and aggregation error under balanced search.

## 3 Preliminaries

In this section, the prerequisites for our proposed method are introduced. We consider an agent learning in a Markov decision process (MDP) represented as $\langle \mathcal{S}, \mathcal{A}, R, \mathcal{T}, \gamma \rangle$, where $\mathcal{S}$ denotes the

Table 1: Some previous state abstraction functions [23, 11, 29].

| Abstractions | Predicate | Transitive |
|---|---|---|
| $\phi_{a^*}$ | $a_1^* = a_2^* \wedge V^*(s_1) = V^*(s_2)$ | yes |
| $\phi_{a^*}^{\varepsilon}$ | $a_1^* = a_2^* \wedge \lvert V^*(s_1) - V^*(s_2) \rvert \leq \varepsilon$ | no |
| $\phi_{Q^*}$ | $\max_a \lvert Q_M^*(s_1, a) - Q_M^*(s_2, a) \rvert = 0$ | yes |
| $\phi_{Q^*}^{\varepsilon}$ | $\max_a \lvert Q_M^*(s_1, a) - Q_M^*(s_2, a) \rvert \leq \varepsilon$ | no |
| $\phi_{Q_d^*}$ | $\forall_a : \left\lceil \frac{Q^*(s_1, a)}{d} \right\rceil = \left\lceil \frac{Q^*(s_2, a)}{d} \right\rceil$ | yes |

state space, $\mathcal{A}$ denotes the action space, $R : \mathcal{S} \times \mathcal{A} \mapsto \mathbb{R}$ denotes the reward function, $\mathcal{T} : \mathcal{S} \times \mathcal{A} \mapsto \mathbb{P}(\mathcal{S})$ denotes the transition model, and $\gamma \in [0, 1]$ is the discount factor. The goal is to learn a policy $\pi : \mathcal{S} \mapsto \mathbb{P}(\mathcal{A})$ that maximizes the long-term expected reward in the MDP.

### 3.1 Monte Carlo Tree Search

MCTS-based algorithms typically involve four stages in the search loop: selection, expansion, simulation, and backpropagation. After $N$ iterations of the search loop, MCTS generates a policy based on the current states. In the selection stage of each iteration, an action is selected by maximizing over UCB. Notably, AlphaZero [2] and MuZero [4], two successful RL algorithms developed based on a variant of Upper Confidence Bound (UCB) [24] called probabilistic upper confidence tree (PUCT) [25], have achieved remarkable results in board games and Atari games. The formula of PUCT is given by Eq. (1):

$$a^k = \text{argmax}_a \, Q(s, a) + c(s) \cdot P(s, a) \frac{\sqrt{\sum_b N(s, b)}}{1 + N(s, a)}, \tag{1}$$

where $k$ is the index of iteration, $Q(s, a)$ denotes the value of action $a$ in state $s$, $c(s)$ is a hyperparameter for balancing the value score with the visiting counts $N(s, a)$, and $P(s, a)$ is the policy prior obtained from neural networks.

### 3.2 State Abstraction in RL

State abstraction methods aggregate similar environment states to compressed descriptions [26], which simplify the state spaces and significantly reduce the computation time [27]. The state abstraction type is formulated as below [10]:

**Definition 3.1.** (State Abstraction Type) A state abstraction type is a set of functions $\phi : \mathcal{S} \mapsto \mathcal{S}_\phi$ related to a fixed predicate on state pairs: $p_M : \mathcal{S} \times \mathcal{S} \mapsto \{0, 1\}$. When function $\phi$ aggregates the state pair $(s_1, s_2)$ in MDP $M$, the predicate $p_M$ must be true: $\phi(s_1) = \phi(s_2) \implies p_M(s_1, s_2)$.

The conditions for state abstraction are usually strict, which may cause insufficient compression in state spaces. Recent studies have shown that transitive state abstraction [23] is computationally efficient and can achieve near-optimal decision-making, which is defined as:

**Definition 3.2.** (Transitive State Abstraction) For a given state abstraction $\phi$ with predicate $p_M$, if $\forall (s_1, s_2, s_3) \in \mathcal{S}$ satisfies $[p_M(s_1, s_2) \wedge p_M(s_2, s_3)] \implies p_M(s_1, s_3)$, the state abstraction $\phi$ is a transitive state abstraction.

Some previous state abstraction functions are shown in Table 1, which aim to abstract similar states in general reinforcement learning methods. For instance, abstraction $\phi_{a^*}$ considers the optimal actions and values of states, which is also widely studied in MCTS-based methods [28].

Transitive state abstraction can reduce the computational cost of finding the smallest abstract state space. As shown in Table 1, most approximate state abstractions are not transitive. However, transitivity is challenging to MCTS-based methods due to the tree structure. In MCTS, a path may contain multiple states, so it is necessary to derive the transitivity of the path from the transitivity of the states. Finding the smallest abstract space in the search tree becomes an NP-hard problem if the state abstraction function lacks transitivity in the path.

# 4  PTSA-MCTS

In this section, we introduce the proposed probability tree state abstraction (*PTSA*) algorithm, which improves the tree search efficiency for MCTS. As described in Figure 1, *PTSA* algorithm improves the search efficiency of MCTS in two aspects: one is reducing the search space by abstracting the original search space, and the other is finding the smallest abstract space efficiently by transitive probability tree state abstraction $\phi_{Q_\alpha^\psi}$. The organization of this section is as follows: Subsection 4.1 gives the formulation of general tree state abstraction. Subsection 4.2 presents our proposed probability tree state abstraction. Subsection 4.3 presents the PTSAZero algorithm, which integrates *PTSA* with Sampled MuZero, and more information can be found in the Appendix. Subsection 4.4 provides the proofs of transitivity in tree structures and bounded aggregation error under balanced exploration.

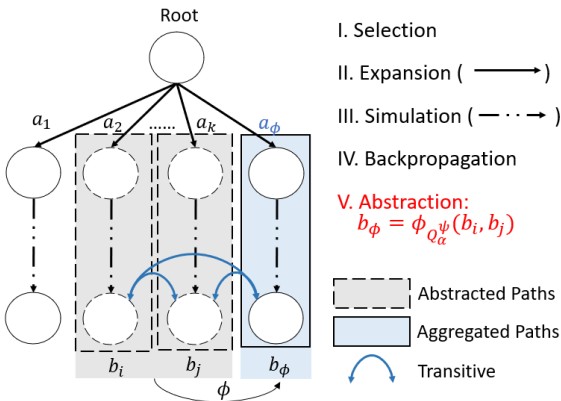

Figure 1: The overview structure of *PTSA* algorithm. The original tree search space in MCTS is mapped into a smaller abstract space efficiently by transitive probability tree state abstraction $\phi_{Q_\alpha^\psi}$.

## 4.1  Tree State Abstraction Formulation

The formulation of our tree state abstraction is provided with a general abstraction operator for the tree structure, which can be utilized with arbitrary state abstraction types. To facilitate the derivation of theorems and properties, the definition of tree state abstraction is given:

**Definition 4.1.** (Tree State Abstraction)

For a given tree, $\mathcal{V}$ and $\mathcal{B}$ denote the node set and path set respectively. A tree state abstraction is a function $\phi : \mathcal{V} \mapsto \mathcal{V}_\phi \ \& \ \mathcal{B} \mapsto \mathcal{B}_\phi$ with node predicate $p_{vM}$ and path predicate $p_{bM}$ on the sibling path pair:

$$
\begin{aligned}
p_{vM} &: \mathcal{V} \times \mathcal{V} \mapsto \{0,1\}; \\
p_{bM} &: \mathcal{B} \times \mathcal{B} \mapsto \{0,1\}.
\end{aligned}
\tag{2}
$$

In the tree structure, a path is a sequence of nodes and a node denotes the representation of the corresponding state. The path and node predicates abstract corresponding states of the path and node respectively. For a given path pair $(b_1, b_2)$ with the same length $l$, the predicate on this path pair can be decomposed as:

$$
p_{bM}(b_1, b_2) = p_{vM}(v_0^{b_1}, v_0^{b_2}) \wedge \cdots \wedge p_{vM}(v_{l-1}^{b_1}, v_{l-1}^{b_2}),
\tag{3}
$$

where $v_{i-1}^b (i = 1, ..., l)$ is the $i$-th node of path $b$ with length $l$. When function $\phi$ aggregates the path pair $(b_1, b_2)$ in MDP $M$, the predicate $p_M$ must be true:

$$
\phi(b_1) = \phi(b_2) \implies p_{bM}(b_1, b_2),
\tag{4}
$$

and path $b_1$ and path $b_2$ belong to the same abstract cluster.

The tree state abstraction is applied to two search paths of equal length that start from the same parent node. This ensures that each node along the paths has a corresponding potential abstracted node, while preserving the Markov property after aggregation. For instance, the path pair $(b_1, b_2)$ from different parent nodes $v_a, v_b$ can not be aggregated, which violates the Markov property in the state transitions $v_a \rightarrow v_0^{b_1} \rightarrow \ldots \rightarrow v_{l-1}^{b_1}$ and $v_b \rightarrow v_0^{b_2} \rightarrow \ldots \rightarrow v_{l-1}^{b_2}$.

Previous studies have proven that tree state abstraction can be an efficient approach to reducing branching factors in MCTS [12, 27, 14]. However, finding the smallest abstract state space is an NP-Hard problem in the previous studies [16]. Following the definition of transitive state abstraction [11], we define the path transitivity in the tree state abstraction formulation:

**Definition 4.2.** (Path Transitivity) For a given tree state abstraction $\phi$ with predicate $p_{bM}$, if $\forall(b_1, b_2, b_3) \in \mathcal{B}$ satisfies $[p_{bM}(b_1, b_2) \land p_{bM}(b_2, b_3)] \implies p_{bM}(b_1, b_3)$, the state abstraction $\phi$ has path transitivity.

The definition of path transitivity extends the equivalence of state abstraction in tree state space and general RL state space. Tree State abstraction for MCTS maps the original path space $\mathcal{B}$ into the abstract path space $\mathcal{B}_\phi$. For non-transitive tree state abstraction, it is necessary to determine whether all possible path pairs belong to the same abstract cluster, and paths may appear repeatedly in path pairs, which requires a massive computation cost to obtain the smallest $\mathcal{B}_\phi$.

## 4.2 Probability Tree State Abstraction

State-of-the-art MCTS-based algorithms utilize neural networks to estimate the value or reward of states. However, hard constraints from previous state abstractions can lead to incorrect aggregation during the early training stage. To reduce the probability of states being mapped to the incorrect abstract space due to bias in network prediction, a novel probability tree state abstraction $\phi_{Q_\alpha^\psi}$ is proposed in this work:

**Definition 4.3.** (Probability Tree State Abstraction $\phi_{Q_\alpha^\psi}$) For a given $\alpha \in [0, 1]$ with node predicate $p_{vM}$ and path predicate $p_{bM}$, the aggregation probability of $\phi_{Q_\alpha^\psi}$ is defined as:

$$\mathbb{P}\{p_{bM}(b_1, b_2) = 1\} \triangleq \mathbb{P}\{\phi_{Q_\alpha^\psi}(b_1) = \phi_{Q_\alpha^\psi}(b_2)\} = 1 - \prod_i^l (1 - \mathbb{P}\{p_{vM}(v_i^{b_1}, v_i^{b_2}) = 1\}); \quad (5)$$

$$\mathbb{P}\{p_{vM}(v_i^{b_1}, v_i^{b_2}) = 1\} \triangleq \alpha(1 - D_{JS}(\mathbb{P}\{Q^\psi(v_i^{b_1}, a)\} \| \mathbb{P}\{Q^\psi(v_i^{b_2}, a)\})). \quad (6)$$

where $\mathbb{P}\{Q^\psi(v, a)\} = \frac{exp(Q^*(v,a))}{\sum_{j \in \mathcal{A}} exp(Q^*(v,j))}$, and $D_{JS}$ is the Jensen-Shannon divergence.

$\phi_{Q_\alpha^\psi}$ encourages nodes that have the same candidate actions with similar value distribution expectations to be aggregated. Using Jensen-Shannon divergence instead of Kullback-Leibler divergence is more advantageous for numerical stability during computation.

## 4.3 Integration with Sampled MuZero

Our proposed *PTSA* algorithm can be integrated with state-of-the-art MCTS-based algorithms. The integration includes two main components: offline learning and online searching. Offline learning involves updating the dynamics, prediction, and value networks by sampling trajectories from a buffer. Online searching involves interacting with the environment to obtain high-quality trajectories, similar to MuZero algorithm. Algorithm1 shows how *PTSA* can be integrated with Sampled MuZero [21] during the searching stage. Compared with the original *Sampled MuZero*, lines 4-12 describe how to collect all searched paths and update the corresponding node values during the multiple iterations. Lines 13-19 describe how tree state abstraction reduces the search space efficiently. Based on Theorem 4.4, abstracting the most recently searched path is enough to find the smallest abstract space for MCTS-based methods. $(\phi(b_i) = \phi(b_s))$ returns a boolean value, where "true" denotes aggregating $b_i$ and $b_s$. This boolean value is determined by calculating the probability $\mathbb{P}(\phi(b_i) = \phi(b_s))$ based on Eq.(5) and Eq.(6).

$S_L$ list records the searched paths in the current search tree. $S_L.delete(b)$ and $S_L.add(b)$ refer to removing and recording path $b$ in $S_L$ respectively. The $pruning(b_j)$ action denotes removing unique nodes of path $b_j$ compared to the other abstracted path in the search tree. $h$ denotes the hidden feature of the original real environment state. Algorithm 1 can be generalized to all tree state abstraction functions by replacing $\phi_{Q_\alpha^\psi}$. The selection, expansion, and backpropagation steps are the same with *Sampled MuZero* [21]. To maintain the balance of $\sum_b N(s, b)$, the visit count of the aggregate node needs to be accumulated into the corresponding state pair. Furthermore, aggregated nodes with different sets of legal actions could lead to unnecessary exploration of invalid or irrelevant parts of the search space, slowing down the search and potentially reducing the quality of the results. In our implementation, we ensure that the aggregated node only expands legal actions for the abstracted state, thus avoiding any negative impacts caused by illegal actions. An undoing aggregation operation has been considered in 1. As the value estimation becomes more accurate, some previously aggregated

**Algorithm 1** PTSAZero

| | | |
|---|---|---|
| 1: | **Input:** Root node $v_0$, dynamics network $\mathbf{d}_\theta$, | 11: Expand child nodes: $v.expand(\pi, h, r)$ |
| | policy network $\mathbf{p}_\theta$, value network $\mathbf{v}_\theta$ | 12: Backpropagation along path $b_s$ |
| 2: | Initialize searched-path list $S_L = \{[v_0]\}$ | 13: $S_L.add(b_s)$ |
| 3: | **for** $n = 0, 1, 2...$ **do** | 14: **for** $b_i \in S_L$ **do** |
| 4: | Reset $v = v_0$, searching branch $b_s = [v]$ | 15: **if** $\phi_{Q_\alpha^\psi}(b_i) = \phi_{Q_\alpha^\psi}(b_s)$ **then** |
| 5: | Selection with **PUCT** | 16: $b_j = \underset{b \in (b_i, b_s)}{\arg\min}(b.V)$ |
| 6: | Add child $v'$ into $b_s$, $v = v'$ | |
| 7: | Update hidden state $h$ and reward $r$: | 17: $v_0.pruning(b_j)$, $S_L.delete(b_j)$ |
| 8: | $h, r = \mathbf{d}_\theta(v.parent.h, a)$ | 18: **end if** |
| 9: | Predict value $V$ and policy $\pi$: | 19: **end for** |
| 10: | $V, \pi = \mathbf{v}_\theta(v.h), \mathbf{p}_\theta(v.h)$ | 20: **end for** |

nodes in the search will no longer be aggregated in the new search. With each new timestep, the value of the search tree nodes is re-evaluated, leading to changes in the following aggregation results.

In Sampled MuZero algorithm, the computational complexity of simulating from the root node can be expressed as $\mathcal{O}(N_s \cdot (\mathcal{O}(S) + \mathcal{O}(D) + \mathcal{O}(P) + \mathcal{O}(V)))$. $N_s$ represents the number of simulations, and $\mathcal{O}(S)$ denotes the computational complexity of the simulation process, which includes selecting children, expanding, and backpropagating. Additionally, $\mathcal{O}(D)$, $\mathcal{O}(P)$, and $\mathcal{O}(V)$ denote the computational complexities of the dynamics network $\mathbf{d}\theta$, policy network $\mathbf{p}\theta$, and value network $\mathbf{v}_\theta$, respectively. The dynamics network predicts the next hidden state $z$ and corresponding reward $r$ based on the current hidden state $h$ and action $a$. In Algorithm 1, the time complexity is given by $\mathcal{O}(N_s \cdot (\log_{|\mathcal{A}|} N_s \cdot c_p + \mathcal{O}(S) + \mathcal{O}(D) + \mathcal{O}(P) + \mathcal{O}(V)))$ under balanced search. The balanced search term $\log_{|\mathcal{A}|} N_s \cdot c_p$ accounts for the exploration of child nodes, where $|\mathcal{A}|$ represents the number of possible actions, and $c_p$ is a constant controlling exploration. Since tree state abstraction reduces the branching factor, our algorithm enhances the efficiency of selecting child nodes with a smaller $N_s$. The specific computational time of different methods can be found in Appendix F.

### 4.4 Theoretical Justification

Following the formulation of the tree state abstractions, the theoretical analysis is conducted from the following perspectives: i) Transitivity; ii) Aggregation error.

#### 4.4.1 Transitivity

To abstract tree paths efficiently, our next result shows the relationship between path transitivity and node transitivity:

**Theorem 4.4.** *For $\forall(v_1, v_2, v_3) \in \mathcal{V}$ and $(b_1, b_2, b_3) \in \mathcal{B}$:*

$$[[p_{bM}(b_1, b_2) \wedge p_{bM}(b_2, b_3)] \implies p_{bM}(b_1, b_3)]] \iff$$
$$[[p_{vM}(v_1, v_2) \wedge p_{vM}(v_2, v_3)] \implies p_{vM}(v_1, v_3)]. \tag{7}$$

**Proof**. See Appendix A.

Theorem 4.4 indicates that path transitivity is a sufficient and necessary condition for node transitivity. Compared with the previous transitive state abstractions, the proposed $\phi_{Q_\alpha^\psi}$ is also transitive for paths as the following proposition given:

**Proposition 4.5.** *The probability of transitivity for $\phi_{Q_\alpha^\psi}$ can be computed as:*

$$\mathbb{P}\{(p_{bM}(b_1, b_2) \wedge p_{bM}(b_2, b_3) \implies p_{bM}(b_1, b_3))\} =$$
$$\mathbb{P}\{\phi_{Q_\alpha^\psi}(b_1) = \phi_{Q_\alpha^\psi}(b_2)\}\mathbb{P}\{\phi_{Q_\alpha^\psi}(b_2) = \phi_{Q_\alpha^\psi}(b_3)\}\mathbb{P}\{\phi_{Q_\alpha^\psi}(b_1) = \phi_{Q_\alpha^\psi}(b_3)\}+ \tag{8}$$
$$(1 - \mathbb{P}\{\phi_{Q_\alpha^\psi}(b_1) = \phi_{Q_\alpha^\psi}(b_3)\})\mathbb{P}\{\phi_{Q_\alpha^\psi}(b_2) = \phi_{Q_\alpha^\psi}(b_3)\})(1 - \mathbb{P}\{\phi_{Q_\alpha^\psi}(b_1) = \phi_{Q_\alpha^\psi}(b_2)\}.$$

**Proof**. See Appendix B.

The computational complexity of computing the smallest possible abstract state space for transitive state abstraction is $\mathcal{O}\left(|\mathcal{S}|^2 \cdot c_p\right)$ [11], where $c_p$ denotes the computational complexity of evaluating for a given state pair.

#### 4.4.2 Aggregation Error Bound

Most approximate state abstractions are non-transitive since their cumulative aggregation value errors are unbounded. We define the aggregation error $E^\phi$ in path set $\mathcal{B}$ as:

$$E^\phi = \sum_{b_1, b_2 \in \mathcal{B}} |V^{\pi_\phi}(b_1) - V^{\pi_\phi}(b_2)|. \tag{9}$$

For instance, consider a commonly used approximate state abstraction $\phi_{Q_\varepsilon^*}$ [23]:

$$p_M(s_1, s_2) \triangleq \max_a |Q_M^*(s_1, a) - Q_M^*(s_2, a)| \le \varepsilon. \tag{10}$$

with the value loss $\mathcal{L}_{\phi_{Q_\varepsilon^*}} = V^*(s) - V^{\pi_{\phi_{Q_\varepsilon^*}}}(s) \le \frac{2\varepsilon R_{max}}{(1-\gamma)^2}$. The aggregation error can be bounded as $\frac{6\varepsilon R_{max}}{(1-\gamma)^2}$ if two paths of length 3 are abstracted.

In Algorithm1, the aggregation times must be less than the current branching factor[2]. The next result extends to the general tree state abstractions in Algorithm1. The path transitivity implies that the abstracted paths present in the current $\mathcal{B}_\phi$ must belong to different abstract clusters, which can give the following theorem:

**Theorem 4.6.** *Considering a general tree state abstraction $\phi$ with a transitive predicate $p(\mathcal{L}_\phi \le \zeta)$, the aggregation error in Alg. 1 under balanced search is bounded as:*

$$E^\phi < \log_{|\mathcal{A}|}(N_s + 1)\zeta. \tag{11}$$

*If predicate $p(\mathcal{L}_\phi \le \zeta)$ is not transitive, the aggregation error is bounded as:*

$$E^\phi < (|\mathcal{A}| - 1)\log_{|\mathcal{A}|}(N_s + 1)\zeta. \tag{12}$$

**Proof.** See Appendix C.

Theorem 4.6 provides a theoretical guarantee for finding the smallest $\mathcal{B}_\phi$ within aggregation error bound in Algorithm1. At the same time, this theorem also points out that the transitivity has an important effect on the tree structure: as the size of the action space $|\mathcal{A}|$ increases, the aggregation error upper bound for transitive tree state abstraction will decrease, while the upper bound for non-transitive state abstraction will increase.

## 5 Experiments

In this section, experiments focus on tree state abstraction for computational efficiency improvement. Firstly, *PTSA* is integrated with state-of-the-art MCTS-based RL algorithms Sampled MuZero and Gumbel MuZero and evaluated on various RL tasks with 10 seeds. The aggregation percentages are also evaluated, which reflects the search space reduction in different tasks. In addition, the comparison between probability tree state abstraction and other state abstraction functions is also conducted.

### 5.1 Results on Performance

To demonstrate the improvement in computational efficiency, the *PTSA* is integrated with Sampled MuZero in Atari and classic control benchmarks, as well as with Gumbel MuZero in the Gomoku benchmark. The performance is compared against several MCTS-based RL algorithms, including (i) MuZero [4], (ii) Sampled MuZero [21], (iii) EfficientZero [5], (iv) Gumbel MuZero [22].

#### 5.1.1 Atari

Atari games are visually complex environments that pose challenges for MCTS-based algorithms[4, 30]. The results for each benchmark and the normalized result are shown in Figure 2, and the shaded intervals represent the standard deviation of the performance across the different random

---

[2]The maximum branching factor is less than $|\mathcal{A}|$ and $|\hat{\mathcal{A}}|$ in *MuZero* and *Sampled MuZero* respectively, where $\hat{\mathcal{A}}$ is the sampled action space

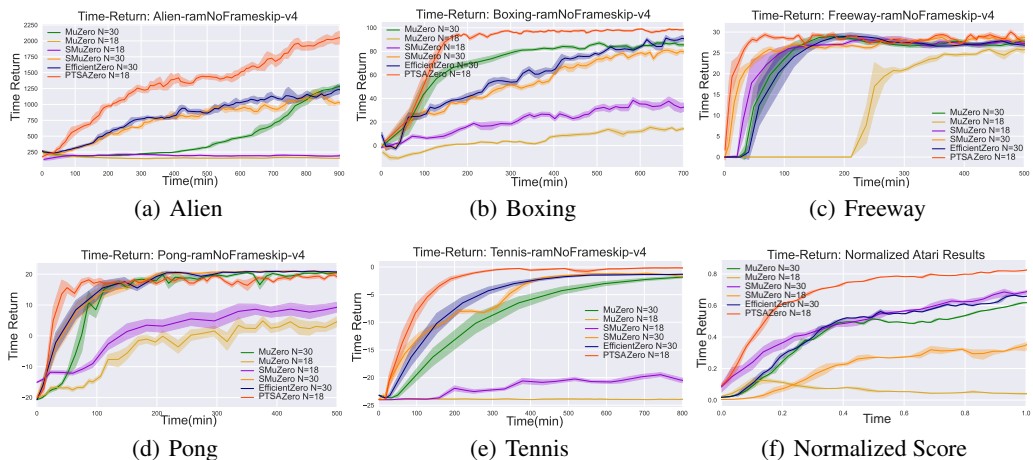

Figure 2: Experiments results on five Atari games (10 seeds) with a normalized score plot. Sampled MuZero with *PTSA* (PTSAZero) is compared with three state-of-the-art MCTS-based methods: MuZero [4], Sampled MuZero (SMuZero) [21], and EfficientZero [5]. The tasks include Alien, Boxing, Freeway, Pong, and Tennis. $N$ denotes the number of simulations. The x-axis is the training time, and the y-axis is the episode return w.r.t training time.

seeds. As Gumbel MuZero does not require large simulations for Atari and control tasks, we only compare its performance in the Gomoku game. When the simulation times $N = 18$, MuZero and Sampled MuZero fail to converge within the maximum training time in some tasks. Since the search space of MCTS is mapped into the abstract space with a smaller branching factor, PTSAZero ($\alpha = 0.7$) can converge rapidly with fewer simulations. Although EfficientZero improves the sampling efficiency of MuZero, the increased complexity of the network entails more time to converge with the same computational resources. EfficientZero is tested with different simulation times, and the best case in time efficiency with $N = 30$ is shown. The normalized score is computed by $s_{norm} = (s_{agent} - s_{min})/(s_{max} - s_{min})$, and the normalized time is computed by $t_{norm} = (t_{agent} - t_{min})/(t_{max} - t_{min})$. The experiment results on Atari benchmarks indicate that Sampled MuZero with *PTSA* can achieve comparable performance with less training time. For MuZero-based algorithms which require massive computational resources and parallel abilities, *PTSA* provides a more efficient method with less computational cost.

### 5.1.2 Control Tasks

In the comparison experiments conducted on Gym benchmarks, including classic control and box2d tasks, certain modifications are made to increase the tree search space in control tasks. Specifically, the action spaces of CartPole-v0 and LunarLander-v2 are discretized into 100 and 36 dimensions, respectively. This discretization of the action space necessitates MCTS-based algorithms to run a larger number of simulations and increases the number of sampled actions in Sampled MuZero.

For both SMuZero and PTSAZero, the number of sampled actions is set to 25 in CartPole-v0 and 12 in LunarLander-v2. Figure 3 demonstrates that PTSAZero exhibits superior computational efficiency compared to other methods. In the LunarLander-v2-36 task, PTSAZero significantly improves the training speed of MuZero by a factor of 3.53. These results highlight the effectiveness of PTSAZero in enhancing the efficiency and performance of MuZero-based algorithms in various control tasks.

### 5.1.3 Gomoku

Gomoku is a classic board game with multi-step search, where the agent is asked to beat an expert opponent on the $15 \times 15$ and $19 \times 19$ boards. Gumbel MuZero replaces heuristic mechanisms in original MCTS algorithms for a smaller number of simulations [22]. In board game tasks, we integrate *PTSA* algorithm with Gumbel MuZero: tree state abstraction $\phi_{Q_\alpha^\psi}$ is utilized to reduce the search space, as described in Algorithm 1. The observation of the Gomoku task is a 2-dimensional matrix describing the distribution of the pieces, and the number of sampled actions for SMuZero,

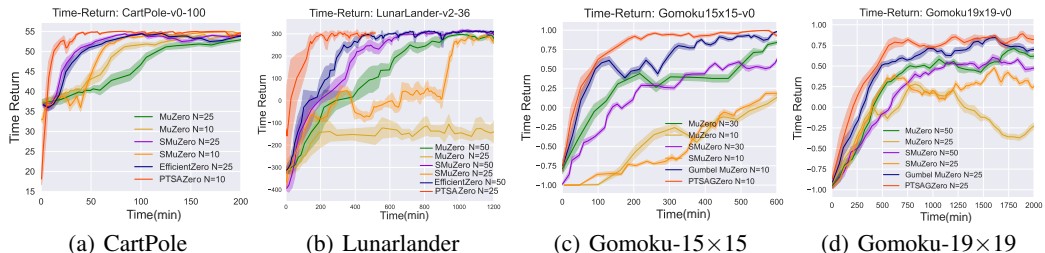

| (a) CartPole | (b) Lunarlander | (c) Gomoku-15×15 | (d) Gomoku-19×19 |

Figure 3: Experiment results of Gym and Gomoku benchmarks (10 seeds). PTSAGZero denotes the Gumbel MuZero with *PTSA* algorithm.

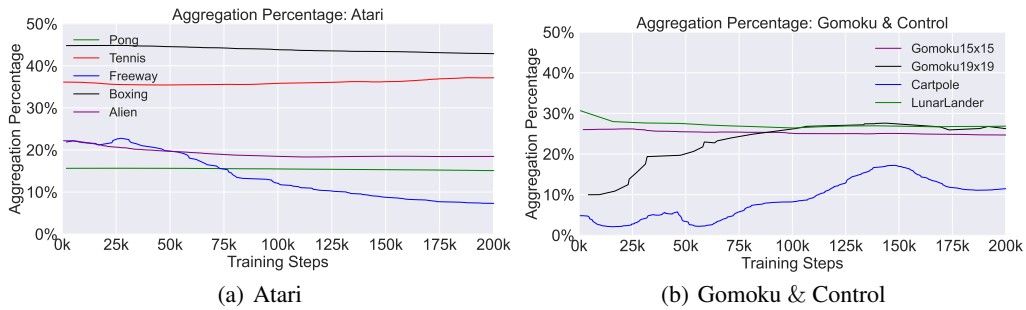

| (a) Atari | (b) Gomoku & Control |

Figure 4: The aggregation percentage on paths during the training process on Atari, Control, and Gomoku tasks varies as the network parameters are updated.

Gumbel MuZero, and PTSAGZero is 30. The training returns of different methods w.r.t. training steps are shown in Figure 3, where a value of 1 represents a win and -1 represents a loss. The results demonstrate that Gumbel MuZero can speed up the training process of Sampled MuZero, and *PTSA* can provide incremental improvement of *Gumbel MuZero*. In the Gomoku-19 × 19 task, despite accelerating the convergence speed of the original algorithms, *PTSA* does not significantly improve the optimal performance of the algorithms. Experimental results also show that MuZero-based algorithms require a larger number of simulations to learn an effective policy due to the increased size of the action space.

## 5.2 Search Space Reduction

To analyze the abstracted tree search space, Figure 4 shows the aggregation percentage (the average proportion of aggregated paths) during the training process of PTSAZero on Atari, Control, and Gomoku tasks. Results demonstrate that tree state abstraction reduces the original branching factors by $10\%$ up to $45\%$. As the network parameters are updated during the training process, the aggregation percentage tends to stabilize. As the training progresses towards convergence, the aggregation percentage decreases from $23.7\%$ to $8.1\%$ in the Freeway task, while in the Gomoku 19x19 task, it increases from $10.1\%$ to $27.8\%$. The trend of the aggregation percentage is influenced by the initialization of network and the initial values for the nodes. The converged aggregation percentage may vary depending on specific algorithm parameters and task environments, and it does not imply a certain range of branching factors for all tasks. Each task has unique characteristics and complexities, which can influence the effectiveness of different abstraction functions. It should be emphasized that a larger reduction in the state space does not guarantee improved training efficiency. While a higher aggregation percentage indicates a greater reduction in the search space, the aggregation quality of the abstractions also determines the impact on training efficiency. Improving the accuracy and quality of tree state abstraction during the training process is a future challenge that deserves further research. Furthermore, since the search space of MCTS is reduced by tree state abstraction, the search depth of PTSAZero is deeper than that of SMuZero with the same number of simulations. More results of search depth can be found in Appendix F.

Table 2: Speedup comparison of PTSAZero with different state abstraction functions, where Abs denotes the different tree state abstraction functions (the notations are shown in Table 1). All state abstraction functions are evaluated under the same number of simulations and sampled actions.

| Abs | Pong | Boxing | Freeway | Tennis | CartPole | Lunarlander | Acrobot | Average |
|---|---|---|---|---|---|---|---|---|
| $\phi_{a^*}$ | 2.35±0.44 | 1.8±0.34 | 2.44±0.46 | 1.59±0.3 | 1.96±0.37 | 3.03±0.57 | 2.11±0.4 | 2.18±0.44 |
| $\phi_{a^*}^{\varepsilon}$ | 2.75±0.52 | 1.77±0.33 | 2.31±0.43 | 1.44±0.27 | 2.0±0.38 | 2.93±0.55 | 1.61±0.30 | 2.12±0.53 |
| $\phi_{Q^*}$ | 2.29±0.43 | **2.11±0.4** | 2.47±0.46 | 1.82±0.34 | 1.64±0.31 | 1.83±0.34 | 1.95±0.37 | 2.00±0.26 |
| $\phi_{Q^*}^{\varepsilon}$ | 3.01±0.59 | 1.95±0.39 | 2.52±0.49 | **2.22±0.44** | 1.74±0.35 | 3.13±0.61 | 2.01±0.40 | 2.37±0.50 |
| $\phi_{Q_d^*}$ | 2.81±0.55 | 1.84±0.37 | 2.45±0.48 | 2.02±0.4 | **2.44±0.48** | 3.19±0.62 | 1.91±0.38 | 2.38±0.46 |
| $\phi_{Q_\alpha^\psi}$ | **3.25±0.19** | 2.00±0.21 | **2.81±0.33** | 1.92±0.17 | 2.14±0.10 | **3.53±0.31** | **2.29±0.35** | **2.56±0.39** |

## 5.3 Comparison with Other State Abstraction Functions

We compare the effectiveness of our proposed probability tree state abstraction with other state abstraction functions on Atari and gym benchmarks by integrating them into *PTSA* algorithm. Referring to the speedup evaluation method in *Gumbel MuZero* [22], Table 2 shows the speedup of PTSAZero with different state abstraction functions compared to the *Sampled MuZero*. The specific usage and properties of other state abstraction functions can be found in previous works [23, 11, 29]. We adjusted hyperparameters for different state abstraction functions and selected the best values ($\epsilon$ and $d$ are set to 0.5 and 0.2, respectively). The results demonstrate that the proposed probability tree state ab-

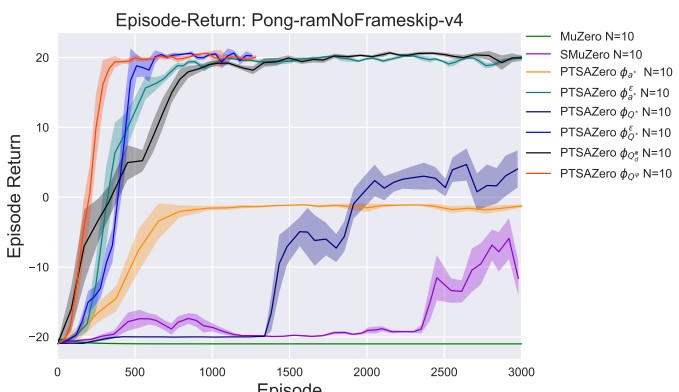

Figure 5: Results of PTSAZero with different state abstraction functions in the Pong task when $N = 10$. Some state abstraction functions cannot accurately abstract the search space with a small number of simulations.

straction $\phi_{Q_\alpha\psi}$ can achieve better speedup performance on average compared to other state abstraction functions. To evaluate the robustness of $\phi_{Q_\alpha^\psi}$, PTSAZero with different state abstraction functions with $N = 10$ is compared in the Pong task. As shown in Figure 5, SMuZero and MuZero can not learn an effective policy, and PTSAZero with $\phi_{Q^*}^{\varepsilon}$ and $\phi_{a^*}$ may aggregate incorrect states, which leads to a decrease in performance. We also find that the state abstraction function $\phi_{a^*}$ and $\phi_{Q^*}$ have strict requirements for the abstract conditions, resulting in a small search space reduction ( 1.5% and 2.8% respectively). Compared with other state abstraction functions, $\phi_{Q_\alpha^\psi}$ can abstract the original search space more accurately and robustly, thus accelerating the training process.

## 6 Conclusion

This paper introduces *PTSA* algorithm for improving the computational efficiency of MCTS-based algorithms. For efficient abstraction in tree search space, we define path transitivity in the formulation of tree state abstraction. Furthermore, we evaluate that the proposed probability tree state abstraction has a better performance compared with previous state abstraction functions. The experimental results demonstrate that *PTSA* can be integrated with state-of-the-art algorithms and achieve comparable performance with $10\% - 45\%$ reduction in tree search space. However, the main limitation of the proposed method is that the parameters of some state abstraction functions need to be manually designed to obtain a more accurate abstract state space. Furthermore, selecting an appropriate state abstraction function based on the characteristics of the state space and transition model is also a potential challenge. Further research will be conducted to address these issues for better performance.

# 7 Acknowledgements

This work was supported by the National Natural Science Foundation of China (Grant No. 92248303 and No. 62373242), the Shanghai Municipal Science and Technology Major Project (Grant No. 2021SHZDZX0102), and the Fundamental Research Funds for the Central Universities.

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
