# A  Path and Node Transitivity

**Theorem A.1.** *For $\forall (v_1, v_2, v_3) \in \mathcal{V}$ and $(b_1, b_2, b_3) \in \mathcal{B}$:*

$$[[p_{bM}(b_1, b_2) \wedge p_{bM}(b_2, b_3)] \implies p_{bM}(b_1, b_3)]] \iff$$
$$[[p_{vM}(v_1, v_2) \wedge p_{vM}(v_2, v_3)] \implies p_{vM}(v_1, v_3)] \tag{13}$$

*Proof.* Consider three paths only contain one node respectively: $b_1 = \{v_1\}, b_2 = \{v_2\}, b_3 = \{v_3\}$. For $(b_1, b_2, b_3) \in \mathcal{B}$:

$$p_{bM}(b_1, b_2) = p_{vM}(v_1, v_2) \tag{14}$$

$$p_{bM}(b_2, b_3) = p_{vM}(v_2, v_3) \tag{15}$$

$$p_{bM}(b_1, b_3) = p_{vM}(v_1, v_3) \tag{16}$$

If the condition is reversed, the equation can also hold. Consider three arbitrary branches (sibling branches of common nodes are omitted): $b_1 = \{v_1, v_2\}, b_2 = \{v_3, v_4\}, b_3 = \{v_5, v_6\}$.

s.t. $p_{vM}(v_1, v_2) \wedge p_{vM}(v_2, v_3) \implies p_{vM}(v_1, v_3)$

According to the definition of branch predicate:

$$p_{bM}(b_1, b_2) = p_{vM}(v_1, v_3) \wedge p_{vM}(v_2, v_4) \tag{17}$$

$$p_{bM}(b_2, b_3) = p_{vM}(v_3, v_5) \wedge p_{vM}(v_4, v_6) \tag{18}$$

$$\begin{aligned} &p_{bM}(b_1, b_2) \wedge p_{bM}(b_2, b_3) \\ &= p_{vM}(v_1, v_3) \wedge p_{vM}(v_2, v_4) \wedge p_{vM}(v_3, v_5) \wedge p_{vM}(v_4, v_6) \\ &= p_{vM}(v_1, v_5) \wedge p_{vM}(v_2, v_6) \\ &= p_{bM}(b_1, b_3) \end{aligned} \tag{19}$$

$\square$

# B  Probability of Transitivity

**Proposition B.1.** *The probability of transitivity for $\phi_{Q_\alpha^\psi}$ can be computed as:*

$$\begin{aligned} &\mathbb{P}\{(p_{bM}(b_1, b_2) \wedge p_{bM}(b_2, b_3) \implies p_{bM}(b_1, b_3))\} = \\ &\mathbb{P}\{\phi_{Q_\alpha^\psi}(b_1) = \phi_{Q_\alpha^\psi}(b_2)\}\mathbb{P}\{\phi_{Q_\alpha^\psi}(b_2) = \phi_{Q_\alpha^\psi}(b_3)\} \\ &\mathbb{P}\{\phi_{Q_\alpha^\psi}(b_1) = \phi_{Q_\alpha^\psi}(b_3)\} + (1 - \mathbb{P}\{\phi_{Q_\alpha^\psi}(b_1) = \phi_{Q_\alpha^\psi}(b_3)\}) \\ &(1 - \mathbb{P}\{\phi_{Q_\alpha^\psi}(b_1) = \phi_{Q_\alpha^\psi}(b_2)\}\mathbb{P}\{\phi_{Q_\alpha^\psi}(b_2) = \phi_{Q_\alpha^\psi}(b_3)\}) \end{aligned} \tag{20}$$

*Proof.* All cases can be divided into two categories:

- $p_{bM}(b_1, b_3) = 1$
- $p_{bM}(b_1, b_3) = 0$

If $p_{bM}(b_1, b_3) = 1$, $p_{bM}(b_1, b_2) = 1$ and $p_{bM}(b_2, b_3) = 1$.

$$\begin{aligned} \mathbb{P}_1 &= \mathbb{P}\{\phi_{Q_\alpha^\psi}(b_1) = \phi_{Q_\alpha^\psi}(b_2)\}\mathbb{P}\{\phi_{Q_\alpha^\psi}(b_2) = \phi_{Q_\alpha^\psi}(b_3)\} \\ &\mathbb{P}\{\phi_{Q_\alpha^\psi}(b_1) = \phi_{Q_\alpha^\psi}(b_3)\} \end{aligned} \tag{21}$$

If $p_{bM}(b_1, b_3) = 0$, $p_{bM}(b_1, b_2) = 0$ or $p_{bM}(b_2, b_3) = 0$.

$$\begin{aligned} \mathbb{P}_2 &= (1 - \mathbb{P}\{\phi_{Q_\alpha^\psi}(b_1) = \phi_{Q_\alpha^\psi}(b_3)\}) \\ &(1 - \mathbb{P}\{\phi_{Q_\alpha^\psi}(b_1) = \phi_{Q_\alpha^\psi}(b_2)\}\mathbb{P}\{\phi_{Q_\alpha^\psi}(b_2) = \phi_{Q_\alpha^\psi}(b_3)\}) \end{aligned} \tag{22}$$

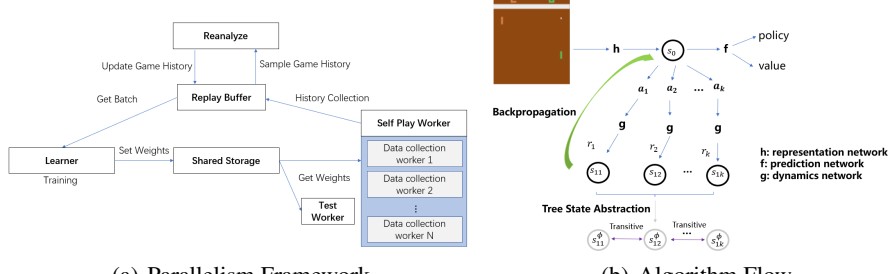

(a) Parallelism Framework         (b) Algorithm Flow

Figure 6: Parallelism framework of PTSAZero implementation and PTSAZero algorithm flow.

$$\mathbb{P}\{(p_{bM}(b_1, b_2) \wedge p_{bM}(b_2, b_3) \implies p_{bM}(b_1, b_3))\} = \mathbb{P}_1 + \mathbb{P}_2 =$$
$$\mathbb{P}\{\phi_{Q_\alpha^\psi}(b_1) = \phi_{Q_\alpha^\psi}(b_2)\}\mathbb{P}\{\phi_{Q_\alpha^\psi}(b_2) = \phi_{Q_\alpha^\psi}(b_3)\}$$
$$\mathbb{P}\{\phi_{Q_\alpha^\psi}(b_1) = \phi_{Q_\alpha^\psi}(b_3)\} + (1 - \mathbb{P}\{\phi_{Q_\alpha^\psi}(b_1) = \phi_{Q_\alpha^\psi}(b_3)\})$$
$$(1 - \mathbb{P}\{\phi_{Q_\alpha^\psi}(b_1) = \phi_{Q_\alpha^\psi}(b_2)\}\mathbb{P}\{\phi_{Q_\alpha^\psi}(b_2) = \phi_{Q_\alpha^\psi}(b_3)\}) \tag{23}$$

□

## C    Aggregation Error Bound of PTSA

**Theorem C.1.** *Considering a general tree state abstraction $\phi$ with a transitive predicate $p(\mathcal{L}_\phi \leq \zeta)$, the aggregation error in Alg. 1 under balanced search is bounded as:*

$$\mathrm{E}^\phi < \log_{|\mathcal{A}|}(N_s + 1)\zeta \tag{24}$$

*If predicate $p(\mathcal{L}_\phi \leq \zeta)$ is not transitive, the aggregation error is bounded as:*

$$\mathrm{E}^\phi < (|\mathcal{A}| - 1)\log_{|\mathcal{A}|}(N_s + 1)\zeta \tag{25}$$

*Proof.* Assuming an action space of size $A$ and expansion of one child node per simulation, the search tree under balanced search in *MuZero* algorithm can be viewed as an $A$-ary tree. The average depth of the tree can be approximated as:

$$D \approx \log_A(N + 1)$$

, where $(N + 1)$ represents the total number of nodes in the search tree, with $+1$ compensating for the root node that is not included in the depth calculation.

Considering transitivity among all searched paths, it is possible to aggregate at most two paths, resulting in a maximum aggregation error equal to the cumulative error of all nodes on these two paths:

$$\mathrm{E}^{\phi_{max}} \leq \log_{|\mathcal{A}|}(N_s + 1)\zeta \tag{26}$$
$$\mathrm{E}^{\phi^r} < \mathrm{E}^{\phi^r_{max}} \leq \log_{|\mathcal{A}|}(N_s + 1)\zeta \tag{27}$$

Considering non-transitivity among all searched paths, all paths should be considered for aggregation, the maximum number of subtrees under the root node in MuZero algorithm is limited by $|\mathcal{A}|$. Therefore, the maximum aggregation error after merging is determined by the cumulative error of all nodes in the largest subtrees under the root node:

$$\mathrm{E}^{\phi^r} < \mathrm{E}^{\phi^r_{max}} \leq (|\mathcal{A}| - 1)\log_{|\mathcal{A}|}(N_s + 1)\zeta \tag{28}$$

## D    Implementation

All experiments are run on Intel Xeon ICX Platinum 8358 and GeForce RTX 3090. The implementation of MuZero is based on the code from **muzero-general** (https://github.com/werner-duvaud/muzero-general) and **model-based-rl** (https://github.com/JimOhman/model-based-rl). The modification of SMuZero has three improvements over MuZero:

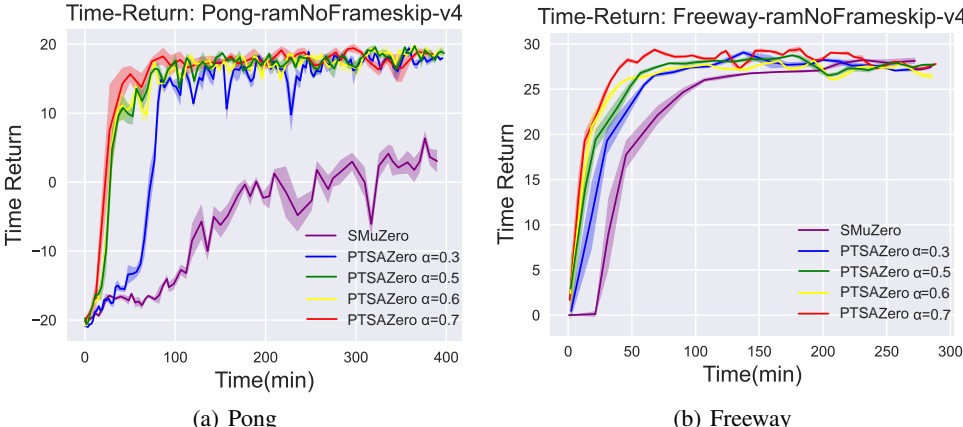

| | |
|:---:|:---:|
| (a) Pong | (b) Freeway |

Figure 7: Experiment results of different parameters in probability state abstraction on Atari benchmarks.

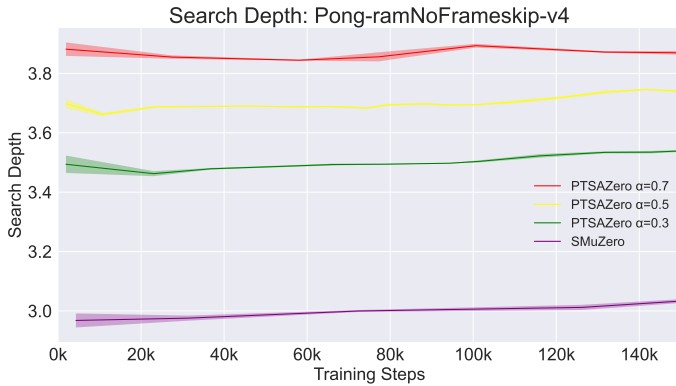

Figure 8: Comparison of average search depth in Pong. The average search depth represents the average path length of the search tree.

- When expanding nodes, the MCTS only considers a set of sampled actions from the original action space, instead of enumerating all actions. The proposal distribution $\beta_p(a|s)$ is based on the policy network, which is consistent with [21].

- The UCB formula does not use the raw prior $\pi$, but instead the sample-based equivalent $\frac{\hat{\pi}}{\pi}$.

- Instead of utilizing the distribution of all actions, the policy is updated on the sampled actions.

The implementation of EfficientZero is based on the code from EfficientZero (https://github.com/YeWR/ EfficientZero). The network structures of all methods are modified as SMuZerO [21] in Atari benchmarks for a fair comparison.

The parallelism implementations of all methods are based on **ray** library [31]. The parallelism framework and algorithm flow of PTSAZero are shown in Figure 6 for better reproduction.

## E Hyperparameters

Conducting experiments on Atari game tasks, the setting of hyperparameters is shown in Table.3. Typically, hyperparameters include learning rate, optimizer, batch size, discount factor, experience replay buffer size, and more. The frame size of the Atari game denotes the pixel size of the observation. In the MuZero-based algorithm, each actor can interact with the environment and collect experience independently, which can increase the amount of experience and reduce the time needed for learning. To ensure equal parallel processing capabilities across all algorithms, we have set the number of

| Parameter | Setting |
|---|---|
| *frame size* | $96 \times 96$ |
| *number of actors* | 7 |
| *max history length* | 500 |
| *visit softmax temperatures* | 1.0,0.5,0.25 |
| *root dirichlet alpha* | 0.25 |
| *root exploration fraction* | 0.25 |
| *pb c base* | 19652 |
| *pb c init* | 1.25 |
| *buffer size* | 10000 |
| *batch size* | 256 |
| *td steps* | 50 |
| *num unroll steps* | 5 |
| *send weights frequency* | 500 |
| *weight sync frequency* | 1000 |
| *discount* | 0.997 |
| *optimizer* | AdamW |
| *lr init* | 0.0008 |

Table 3: Specific parameters in Atari benchmark.

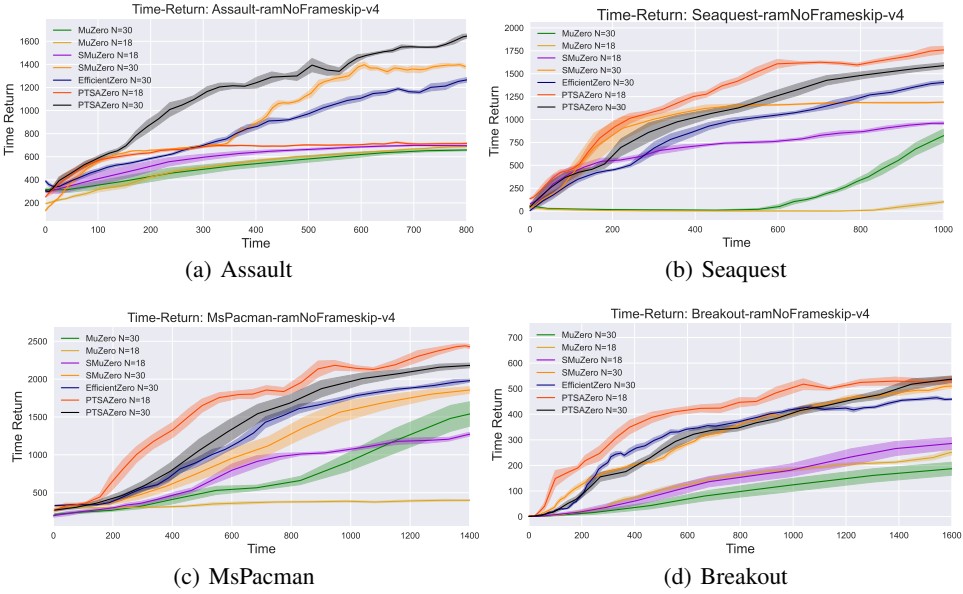

(a) Assault      (b) Seaquest

(c) MsPacman      (d) Breakout

Figure 9: More experimental results on Atari benchmarks.

actors to 7 for each method. This uniform setting helps to ensure that each method can effectively utilize parallel processing resources.

## F  Additional Experiments

For a clear numerical comparison, Table 4 shows the average computation time of collecting 1k frames with different simulation times on Atari benchmarks. Compared to other algorithms, PTSA introduces an acceptable decrease in trajectory collection efficiency (less than $8\%$ on average), which results in a significant reduction in the whole training time. Additionally, we compare different $\alpha$ in probability tree state abstraction, and results are shown in Figure 7. Results demonstrate that the algorithm's convergence speed improves as the parameter $\alpha$ increases.

Table 4: Average computation time (seconds) of collecting 1k frames in Atari benchmarks. Box. denotes Boxing, Free. denotes Freeway, Ten. denotes Tennis, Break. denotes Breakout, MsP. denotes MsPacman, and Sea. denotes Seaquest tasks respectively. Ave. denotes Average computation time.

| Methods | Box. | Free. | Pong | Alien | Ten. | Assault | Break. | MsP. | Sea. | Ave. |
|---------|------|-------|------|-------|------|---------|--------|------|------|------|
| *MuZero N=30* | 6.31 | 3.47 | 4.56 | 8.85 | 3.86 | 3.41 | 3.24 | 3.43 | 3.18 | 4.48 |
| *SMuZero N=30* | 6.89 | 4.43 | 4.44 | 8.89 | 4.02 | 3.47 | 3.33 | 3.50 | 3.35 | 4.70 |
| *PTSAZero N=30* | 6.74 | 3.85 | 4.81 | 9.04 | 4.11 | 3.58 | 3.50 | 3.63 | 3.37 | 4.74 |
| *MuZero N=18* | 3.06 | 2.39 | 2.46 | 3.49 | 3.42 | 1.71 | 1.61 | 1.92 | 1.69 | 2.42 |
| *SMuZero N=18* | 4.12 | 2.41 | 2.11 | 3.07 | 3.45 | 1.76 | 1.97 | 1.98 | 1.72 | 2.51 |
| *PTSAZero N=18* | 3.41 | 1.96 | 3.16 | 3.37 | 3.55 | 1.85 | 1.86 | 2.05 | 1.84 | 2.56 |

Moreover, the comparison results of average search depth between SMuZero and PTSAZero with different $\alpha$ are shown in Figure 8. Since the search space of MCTS is reduced by tree state abstraction, the search depth of PTSAZero is deeper than that of SMuZero with same number of simulations.