# OpenReview forum: "Accelerating Monte Carlo Tree Search with Probability Tree State Abstraction"
_NeurIPS.cc/2023/Conference — NeurIPS 2023 poster_

### Official Review · Reviewer_Yoim · 2023-06-27

**Soundness:** 3 good
**Presentation:** 3 good
**Contribution:** 4 excellent
**Rating:** 7
**Confidence:** 4

**Summary:**

This paper presents a novel approach called Probability Tree State Abstraction (PTSA) to improve the efficiency of Monte Carlo Tree Search (MCTS) algorithms, which have shown remarkable performance in challenging tasks. The computational complexity of MCTS algorithms is influenced by the size of the search space, and the proposed PTSA algorithm aims to address this issue. The algorithm introduces a general tree state abstraction with path transitivity, which helps in reducing the number of mistakes during the aggregation step. The theoretical guarantees of transitivity and aggregation error bound are also provided. The PTSA algorithm is integrated with state-of-the-art MCTS-based algorithms, including Sampled MuZero and Gumbel MuZero, and experimental results on various tasks demonstrate its effectiveness. The PTSA algorithm accelerates the training process of these algorithms, achieving a search space reduction of 10% to 45%.


**Strengths:**

1. The approach of aggregation considers the entire path, not only a state, is novel and unique.

2. The PTSA algorithm presented in this paper can be applied with any other state abstraction functions mentioned in previous studies, in a general way.

3. The paper provides extensive experimental data. It includes environments such as Atari games, as well as tasks with continuous action spaces like CartPole and LunarLander, and board games like Gomoku. The rich variety of experimental environments demonstrates the effectiveness of the proposed method across various tasks.

4. Integrate PTSA with state-of-the-art algorithms can achieve comparable performance with smaller branching factors. In other words, PTSA provides a more efficient method with less computational cost.


**Weaknesses:**

1. The meaning of probability in PTSA (in Definition 4.3) is not well-defined and requires further clarification. This will be addressed in the Questions section below.

2. There are some errors in the proofs presented. This will be discussed in detail in the Questions section as well.


**Questions:**

1. Why does $v_0.pruning$ do in line 17 in Algorithm 1? Any difference from $S_L.delete(b_j)$.

2. What role does the probability $\mathbb{P}$ in Definition 4.3 play in Algorithm 1?And, how to calculate $\phi$ in line 15 in Algorithm 1? In other words, when does $\phi(b_i)=\phi(b_s)$ hold true? Are both related?

3. In your paper, you mentioned a previous work titled "Monte Carlo Tree Search with Iteratively Refining State Abstractions." That method directly calculates the distance between states and performs aggregation if the distance, denoted as $d(s_1, s_2)$, is below a threshold. This approach differs from the method proposed in your paper, but both aim to reduce the branching factor of MCTS. Have you conducted any experiments comparing your method with the approach mentioned above? I couldn't find any analysis of that method in Table 1 or the experimental section below. Some insight into the reason for this omission should be provided.

4. This paper mentioned “reduce the computation time” with abstraction. My question (or curiosity) is how much overhead the checking operations (in Lines 14 and 15) incur. Note that in line 207 there is a time complexity which is required to be described in more detail, like $\log N_s$.

5. Equation (19) in the appendix is written incorrectly. Need to be fixed. For example, the final $p_{bM}(b_2, b_3)$ should be $p_{bM}(b_1, b_3)$. Also fix some other wrong indices in (19).


**Limitations:**

N.A.

---

> ### Author Rebuttal · Authors · 2023-08-07
>
> Thanks for your uplifting review and valuable feedback. We appreciate your comments and would like to address them below.
>
> **Questions:**
>
> 1. We apologize for not providing a clear explanation of the pruning/delete/add actions in the paper. We have provided more detailed explanations of the actions and notation used in Algorithm 1:
> $S_L $ is a list that records the searched paths in the current search tree. $S_L.delete(b)$ and $S_L.add(b)$ refer to removing and recording path $b$ in $S_L$ respectively. The $pruning(b_j)$ action denotes removing unique nodes of path $b_j$ compared to the other abstracted path in the search tree.
>
>   2. We sincerely apologize for any misunderstanding caused by the simplification of the abstraction decision in Algo. 1. $(\phi(b\_i)=\phi(b\_s))$ returns a boolean value, where "true" denotes aggregating $b\_i$ and $b\_s$. This boolean value is determined by calculating the probability $\mathbb{p}(\phi(b\_i)=\phi(b\_s))$ based on Equations (5) and (6). In the practical implementation, once the probability is computed, a random number (0~1) is generated and compared to the probability. If the random number is less than the probability, $\phi(b\_i)=\phi(b\_s)$ holds true. We will provide a clearer explanation of this issue in the final version.
>
>  3. The work presented in "Monte Carlo Tree Search with Iteratively Refining State Abstractions" makes significant contributions by introducing an alternative approach called "abstraction refining" to replace progressive widening in MCTS. Their experiments primarily compare the "abstraction refining" method with progressive widening. As "abstraction refining" provides an improvement during node selection and expansion, it does not conflict with our method of performing abstraction after completing the backpropagation. Additionally, due to the challenge of accurately calculating the distance between hidden states, there is a lack of a clear implementation for integrating the "abstraction refining" method with MuZero algorithm. Hence, this comparison is difficult to be conducted in our experiments.
> Furthermore, Table 1 focuses on describing the state abstraction functions used in previous model-free algorithms, which is why the "abstraction refining" method is not included in Table 1. In future work, we plan to study the impact of incorporating the criterion $d(s_1, s_2)$ in our algorithm.
>
> 4. According to your suggestion, we have conducted more experiments including 9 Atari games to further discuss the limitation of the added computational complexity. Experimental results (shown in Table 1 of the uploaded PDF) demonstrate that PTSA introduces an acceptable decrease in trajectory collection efficiency (less than 8% on average), which results in a significant reduction in the whole training time.
>
> 5. Thanks for your careful review. We have corrected the typos in Appendix Equation (19):
>   \begin{equation}
>   p_{bM}\left(b_{1}, b_{2}\right) \wedge p_{bM}\left(b_{2}, b_{3}\right)
>   = p_{vM}\left(v_{1}, v_{3}\right) \wedge p_{vM}\left(v_{2}, v_{4}\right) \wedge p_{vM}\left(v_{3},
>  v_{5}\right) \wedge p_{vM}\left(v_{4}, v_{6}\right)
>   = p_{vM}\left(v_{1}, v_{5}\right) \wedge p_{vM}\left(v_{2}, v_{6}\right) \\
>   = p_{bM}\left(b_{1}, b_{3}\right).
>   \end{equation}
>   We appreciate your feedback and support.

---

### Official Review · Reviewer_Bu9r · 2023-07-03

**Soundness:** 2 fair
**Presentation:** 3 good
**Contribution:** 2 fair
**Rating:** 7
**Confidence:** 4

**Summary:**

This paper proposed a novel search algorithm, PTSA, to improve the search efficiency of MCTS. Empirical results show that PTSA can be integrated with Sampled MuZero and Gumbel MuZero and can reduce the original branching factor by 10% up to 45%.

**Strengths:**

The proposed PTSA algorithm can reduce the branching factor of MCTS and improve the computational efficiency of MCTS-based algorithms. The authors also provide both theoretical and empirical analyses.

**Weaknesses:**

* The author claims that the proposed method can reduce the branching factor by 10% up to 45%. However, the result is based on only five Atari games. Based on Figure 3, the aggregation percentage varies across different Atari games. Can these five games represent all Atari-57 games? It would be more convincing to run more Atari games to support the claims.

* Moreover, it is unclear for the aggregation percentage on control tasks and Gomoku experiments. Without these experiments, it is inappropriate to claim “reduce branching factor by 10% up to 45%”.

* The time complexity of the proposed approach is higher than the original MCTS. It is unclear whether PTSAZero will still improve its efficiency when running under a larger simulation number. Currently, the authors only run “PTSAZero N=18” in Atari experiments. Will “PTSAZero N=30” perform better than “PTSAZero N=18”?

* Besides, in the board games such as Gomoku or Go, it is common to run large simulation numbers such as N=400 or N=800 during evaluation. It would be better to provide additional experiments/analyses to demonstrate the scale-up ability for PTSAZero. For example, providing the aggregation percentage/time usage/strength when using different simulation numbers.

**Questions:**

* In Algorithm 1, line 15, if $b_i$ and $b_s$ have different lengths, will their $\phi_{Q_{\alpha}^{psi}}(b)$ be different? In addition, what is the definition for $\phi_{Q_{\alpha}^{psi}}(b)$? Definition 4.3 only shows the probability.
* In Algorithm 1, line 17, $v_0$ is root node and $b_j$ is a selection path. what does $v_0$.prunning($b_j$) mean?
* In Figure 2, will PTSA get better performance when using a larger simulation (N=30)? Current experiments only used N=18. It would be better to add another experiment with a larger simulation to show the scale-up ability of PTSA.
* In the Gomoku experiment, what does the expert opponent stand for? How many simulations are used in the Gomoku evaluation? As Gomoku is a two-player game, why not compare PTSAZero to other methods directly?
* line 302: “The winning rates of different methods w.r.t. training time are shown in Figure 4”. Should the range of the win rate be between 0 and 1 in Figure 4?
* In Figure 3, it seems that the aggregation percentage varies across different Atari games. Which type of game may have a higher aggregation percentage? Why do you choose these games? Can these five games represent Atari-57 games? Do you have more experiments on other Atari games?
* In Atari experiments, “As Gumbel MuZero does not require large simulations for Atari and control tasks”. In fact, Gumbel MuZero improves training efficiency by only using N=2 in Pacman, and the result is comparable to N=50. It would be more convincing to add additional experiments to compare the training efficiency between “Gumbel MuZero N=2” and “PTSAGZero N=2“ in Atari experiments.
* In Figure 2 (f),  the label of the green curve is “MuZero N=50”, should it be “MuZero N=30”?
* Line 17, typo: Muzero -> MuZero.
* Figure 2, typo: state-of-art -> state-of-the-art.
* Figure 3 is shown after Figure 4. Please fix the order of these figures.

**Limitations:**

The authors have addressed the limitations in the paper.

---

> ### Author Rebuttal · Authors · 2023-08-07
>
> Thank you for the detailed review and the explicitly listed concerns. We hope that our answers will help to resolve the concerns.
>
> **Weaknesses**:
>
> 1. We appreciate your feedback and acknowledge your concern regarding the limited number of Atari games used in our evaluation. We have conducted more Atari experiments (Assault, Seaquest, Breakout, and MsPacman tasks). The experimental results can be found in Figure 1 (in the uploaded PDF), which demonstrate the effectiveness of our algorithm in training acceleration.
> 2. We would like to emphasize that our method improves the training efficiency with search space reduction. In our experiments including Gomoku, Control, and Atari tasks, we observe a 10%-45% search space reduction. As shown in Figure 2 (in the uploaded PDF), we provide additional evidence of reduced branching factors in more tasks to support this statement. We will revise our claim to eliminate this ambiguity.
> 3. Figure 1 in the uploaded PDF shows the training results of each method with $N \in \\{ 18,30\\}$, indicating that PTSA exhibits superior training efficiency compared to other algorithms with the same simulations. This result demonstrates that the PTSA also improves the training efficiency with larger simulations.
> We also compare the performance of PTSA N=30 and PTSA N=18. The results show that PTSA N=18 achieves comparable performance as PTSA N=30  with less training time. The presentation of results for N=18 in the paper aims to highlight how PTSA enables MCTS-based algorithms to achieve comparable performance even at smaller N values, thereby enhancing training efficiency.
>
> 4. We appreciate your suggestion. To compare the effectiveness and aggregation percentage of PTSAZero with different numbers of simulations ($N$), we have conducted additional experiments as shown in Figure 2&3 (in the uploaded PDF). We observe that increasing the number of simulations does not significantly affect the aggregation percentage for the same environment and state abstraction function.
>
> **Questions:**
>
> 1.  The two abstracted paths are required equal in length, which aligns with the mapping principle of state abstraction theory. Given two paths to be abstracted, the mapping principle denotes that two abstracted nodes at the same depth should be mapped into the same abstracted state. If two paths are of different lengths, their abstracted path lengths will also be different, which conflicts with the mapping principle.
> Similar to other state abstraction functions, the $\phi_{Q^{\psi}_{\alpha}}(b)$ function maps a given path to a path in the abstract space. In Section 4.3, we define a method to compute the probability of two paths being mapped to the same abstract path under $\phi\_{Q^{\psi}\_{\alpha}}(b)$. We will provide a clearer explanation in the final version.
> 2. We apologize for not providing a comprehensive explanation of the pruning operation in the paper. We have provided more detailed explanations of the actions and notation used in Algorithm 1, including the pruning/delete/add operations. $S_L$ is a list that records the searched paths in the current search tree. $S_L.delete(b)$ and $S_L.add(b)$ refer to removing and recording path $b$ in $S_L$ respectively. The $pruning(b_j)$ action denotes removing unique nodes of path $b_j$ compared to the other abstracted path in the search tree.
> 3. According to your suggestions, we have conducted additional experiments. The results can be found in Figure 1 (in the uploaded PDF), while the analysis is given in response to the weakness.
>
> 4. Two methods including pitting them against each other and employing the same rule-based agent as the opponent are both commonly used to evaluate AI agents in board games. Using the same rule-based agent as the opponent is sufficient for a fair comparison in our experiments, which is commonly utilized in previous works [1,2,3]. Additionally, to maintain consistency, we use the same number of simulations for evaluation as during training.
>
>   5. We apologize for the incorrect use of the term "win rate." It should have been referred to as the "return" in the figure, where a value of 1 represents a win and -1 represents a loss during evaluation. We will rectify this typo in the final version.
>
>   6. You have raised an intriguing point that the characteristics of the state space in different tasks and the state abstraction function are the key factors influencing the reduction effect. This presents an important direction for future research.
> We have conducted more Atari experiments, and the corresponding results can be found in the uploaded PDF. 9 Atari games are utilized in our experiment, including Pong, Alien, Breakout, etc., which are commonly used tasks in the previous works.
>
>   7. Our motivation is to enhance MCTS efficiency by reducing the branching factor. However, when N is very small, the advantages of the PTSA algorithm diminish due to the extremely limited search space. Considering that N=2 already results in a significantly small branching factor, it is unnecessary to evaluate the performance in such a small N scenario.
>
> 8,9,10,11 We will correct these typos and mistakes in the final version. Thank you very much for pointing out these issues, and we will address these issues and improve readability in the final version.
>
> [1] Spending thinking time wisely: Accelerating MCTS with virtual expansions
>
> [2] Action guidance with MCTS for deep reinforcement learning
>
> [3] Opponent modeling based on action table for MCTS-based fighting game AI

---

> > ### Comment · Reviewer_Bu9r · 2023-08-16
> >
> > Thank you for the detailed responses and additional experiments. Do you have experiments on using large simulation numbers (such as N=400 or N=800) on Gomoku? Since PTSA significantly reduces the branching factors, it would be great to see more reduction rates when using large simulations.

---

> > > ### Author Response · Authors · 2023-08-20
> > >
> > > Thanks for your response. Due to the several days required for training with N=400 and N=800, the results were not immediately presented in the  uploaded PDF. The following table presents the average reduction rate with different simulations in the Gomoku-19$\times$19 task, where the reduction rate increases with the increase in the number of simulations.
> > >
> > > | Simulations| reduction rate|
> > > |------------------|------------|
> > > | N=25               | 28.3%         |
> > > | N=100             | 33.4%         |
> > > | N=400             | 37.7%         |
> > > | N=800             | 40.3%         |

---

> > > > ### Comment · Reviewer_Bu9r · 2023-08-21
> > > >
> > > > The author's additional experimental results have addressed my concerns, and I truly appreciate it. For further clarification of this paper, I will suggest including these experiments in the final version. I have also increased my score.

---

### Official Review · Reviewer_GnyW · 2023-07-07

**Soundness:** 3 good
**Presentation:** 2 fair
**Contribution:** 3 good
**Rating:** 7
**Confidence:** 2

**Summary:**

The paper proposes a novel tree state abstraction function (PTSA) for use during MCTS. The primary contributions of the paper are algorithmic and empirical. The key idea involves aggregating paths in the tree if their Q-values (as probabilities) along the path closely match an existing path with the same parent node. An analysis of the abstraction quality and error bounds are included. Experiments on Atari and Gym environments show that a recent MCTS variant leveraging PTSA outperforms a number of strong baselines.

UPDATE: I thank the authors for their detailed response. After reading the other reviews and comments, I'm more inclined to recommend acceptance and have updated my score to reflect that.

**Strengths:**

+ The paper tackles an important problem of accelerating MCTS search. It does so using tree state abstraction. The approach is intuitively clear and is also supported by analysis.

+ The paper proposes a novel tree state abstraction function based on path transitivity. The abstraction function is based on the difference in the Q values of the nodes (converted to probabilities) in the path. Although important implementation details are not clear to me, the intuition that abstracting entire paths accelerates search makes sense as does the abstraction of only the most recent path during search leading to smaller trees during online search. The paper is accompanied by analysis showing the correctness of the approach and an error bound under certain conditions. Overall, the algorithm seems to have high novelty.

+ The experiments are conducted on a number of Atari and Gym environments. Sampled MuZero with the proposed abstraction (PTSA) outperforms a number of strong baselines by a significant margin. The implementation seems to work very well. This seems to be a new state of the art in state abstraction for modern MCTS variants.

**Weaknesses:**

- The approach is intuitively clear and seems to perform well empirically, which increases confidence. However, I found the description of the implementation details of Algorithm 1 difficult to follow. Please consider including a more careful description of the implementation in Section 4. The issue is exacerbated by the absence of code. This is currently preventing me from giving the paper a higher score.
  - For example, the actual implementation of the algorithm in L15 of Algorithm 1 is unclear to me. I expect it to involve Eq 5 with some value of $\alpha$ like 0.7. But Eq 5 returns a real-valued prob estimate for a path pair (b_i, b_s). How is that turned into a boolean value (True / False) in L15? It's probably not real-valued equality. This is a key detail so please explain.
  - There are a number of other implementation details that are difficult to find or missing. See the questions for examples.

- Given that the primary contribution is algorithmic and empirical, I'd have hoped to see the source code included. Reproducibility is going to be challenging without it and since this paper seems to establish a new state of the art, I'd encourage the authors to release their code.

**Questions:**

- I had a number of questions about the exact implementation
  - What exactly is the implementation of the branching condition of L15 of Algorithm 1? How does it relate to Eq 5 and 6 which is defined as a function taking two inputs (b_i, b_j) and returning a probability?
  - What exactly is learned during offline learning vs online search? $d, v, p$ are likely learned offline. What about the abstraction function $\phi$? This seems online to me. Correct?
  - What is $l$ set to in the implementation? How does it value impact performance?
  - What is the implementation of the pruning function in L17 of Algorithm 1?
  - How are the legal actions for the abstracted state computed?
  - What is the size of $S_L$? How was it chosen? How does varying it affect performance?

- As described in L346-349, there seem to be a number of choices for the designer to make. These are not clear to me at the moment besides the obvious ones (e.g., $\alpha, N$). Please enumerate what exactly needs to be hand-designed or set manually for a given domain and what can be used off-the-shelf.

- Is there a reason to not include code? Will code be included in the final version?

**Limitations:**

Yes

---

> ### Author Rebuttal · Authors · 2023-08-07
>
> Thanks for your valuable feedback. We address each comment and concern below.
>
> **Weaknesses:**
>
> 1. We appreciate your feedback and will provide a more careful description of the implementation details in the final version. We have provided more detailed explanations of the pruning/delete operation used in Algorithm 1. Detailed modifications can be found in the general response.
> 2. We will release our code upon acceptance, and conduct further research to explore how different state abstraction functions can be better tailored to fit the characteristics of the environment. We hope that this research will help to enhance the performance and applicability of our approach in a wider range of applications.
>
> **Questions:**
>
> 1. Questions about the exact implementation：
>
>     1. We sincerely apologize for any misunderstanding caused by the simplification of the abstraction decision in Algo. 1. $(\phi(b\_i)=\phi(b\_s))$ returns a boolean value, where "true" denotes aggregating $b\_i$ and $b\_s$. This boolean value is determined by calculating the probability $\mathbb{p}(\phi(b\_i)=\phi(b\_s))$ based on Equations (5) and (6). In the practical implementation, once the probability is computed, a random number (0~1) is generated and compared to the probability. If the random number is less than the probability, $\phi(b\_i)=\phi(b\_s)$ holds true. We will provide a clearer explanation of this issue in the final version.
>
>     2. Yes, your understanding is correct. Offline learning involves updating the dynamics, prediction, and value networks by sampling trajectories from a buffer. Online searching involves interacting with the environment to obtain high-quality trajectories, similar to MuZero algorithm. However, due to space limitations, Algorithm 1 in the paper only provides the MCTS search process, which may cause confusion for readers. We would like to clarify this issue in the final version.
>
>     3. The parameter $l$ is determined by the number of non-shared nodes between paths $b_i$ and $b_j$. A larger value of $l$ indicates that more nodes need to be evaluated to determine if they satisfy the aggregation condition. When $l=1$, path aggregation will degrade to node aggregation.
>
>     4. We apologize for not providing a clear explanation of the pruning/delete/add actions in the paper. We have provided more detailed explanations of the actions and notation used in Algorithm 1:  $S\_L $ is a list that records the searched paths in the current search tree. $S\_L.delete(b)$ and $S\_L.add(b)$ refer to removing and recording path $b$ in $S\_L$ respectively. The $pruning(b\_j)$ action denotes removing unique nodes of path $b\_j$ compared to the other abstracted path in the search tree.
>
>     5. The process of obtaining legal actions is similar to MuZero, which stores hidden states rather than real states. Therefore, the legal actions at the root node are obtained based on the corresponding real states. For example, in a board game, the legal actions represent the legal moves available in the current real state.
>
>     6. $S\_L $ is a list that records the searched paths in the current search tree. In algorithms such as MuZero, SMuZero, and EfficientZero, the size of $S\_L$ equals the number of simulations conducted. In our proposed algorithm, the size of $S\_L$ is adjusted by the number of simulations subtracting the number of path aggregations. A larger $S\_L$ indicates a larger tree search space, which may lead to inefficient exploration.
>
> 2. We would like to resolve your concers from the following two aspects:
>
>     a. State abstraction functions need to be selected based on different environments: For example, $\phi_{a^*}$ requires the values of two states to be exactly equal and have consistent optimal actions. Such abstraction conditions can be inefficient for some tasks, such as Atari, where two states may have similar but not equal values. For different tasks, it may be necessary to choose suitable state abstraction functions to achieve better abstraction performance.
>
>    b. The parameters of the state abstraction function require manual tuning: Some parameters of the state abstraction function need to be adjusted manually based on the characteristics of the environment. For instance, the parameter $d$ in $\phi\_{Q^*\_d}$ typically ranges from 0 to 1. If the Q-value distribution in the environment is relatively smooth, a smaller value for $d$ (e.g., 0.2) is generally preferred. The parameter $\epsilon$ in $\phi^{\epsilon}\_{Q^*}$ is adjusted according to the range of the reward function. When the reward function has a larger range, the value of $\epsilon$ tends to be relatively larger as well. Furthermore, you mentioned some parameters used in MCTS-based algorithms, such as simulations $N$. These parameters can be informed by previous experiences with MCTS work. For example, in Atari tasks, a common choice for the number of sampled actions $K=6$.
>
>     Studying how to select suitable state abstraction functions and fine-tune the parameters accordingly can be a direction for future research. We hope these responses can resolve your concerns.
>
> 3. We will release our code upon acceptance, and provide the training instructions along with the corresponding random seed settings.

---

> > ### Comment · Reviewer_GnyW · 2023-08-19
> >
> > I thank the authors for their detailed response. After reading the other reviews and comments, I'm more inclined to recommend acceptance and have updated my score to reflect that.

---

### Official Review · Reviewer_GJR1 · 2023-07-17

**Soundness:** 3 good
**Presentation:** 3 good
**Contribution:** 2 fair
**Rating:** 4
**Confidence:** 5

**Summary:**

To accelerate MCTS, the paper proposed a novel probability tree state abstraction (PTSA) algorithm to improve the search efficiency of MCTS.
They define states that are similar by using path transitivity and claim that such a method can have fewer mistakes. According to the results of Atari and Gomoku, the method can be 10% ~ 45% faster.


**Strengths:**

1. The method provides some theoretical guarantee.
2. The experiments take place in many environments.
3. The ablation studies have tried many abstraction functions.


**Weaknesses:**

1. The intuition of the paper is weird.  The method required of all states on the paths needs to be similar. However, there are two problems here. First, the V value might be more incorrect at the beginning. Second, even if the V value is correct for the whole path, this method reduces the chance of pruning more nodes. For example, in Atari, agents can reach the exact same state with different paths. Since the environment is MDP, we should merge those two states.

2. It is unknown for the performance when the simulation is higher. The abstract error normally increases when the simulation increase. The method might delete some good paths that can only be identified after numerous simulations.




**Questions:**

1. How do you prune a path from a tree? What will happen to those branches that are on the path?
2. Have you considered abstraction functions that also require the best action should be the same[1]?
[1] Are AlphaZero-like Agents Robust to Adversarial Perturbations?

**Limitations:**

Stated in the weakness.

---

> ### Author Rebuttal · Authors · 2023-08-07
>
> Thanks for your helpful comments. We hope that our answers will help to resolve your concerns.
>
> **Weaknesses:**
>
> 1. For the first problem, during the early stage of training, inaccurate estimation of the V values may prevent state abstraction methods from correctly aggregating states, which is a common issue for previous state abstraction methods. Addressing this issue is indeed a significant contribution of this paper. PTSA provides improved fault tolerance compared to previous abstraction functions by probabilistically avoiding incorrect aggregations, especially during the early stage of training, which is consistent with the results presented in Section 5.2 of our experiments.
>
>     For the second problem, the aggregation cannot be applied to all the similar or same states in the tree search space. For instance, given two paths with the same start node: $b_1=(v_0,v_1,v_2,\cdots, v_n)$ and $b_2=(v_0,v'_1,v'_2,\cdots, v'_n)$. $v_n$ and $v'_n$ represent similar or the same states. Assume  $v_n$ and $v'\_n$ are aggregated into the same node. However, this operation leads to a ring structure, which conflicts with the MCTS tree structure. Intuitively, this issue can be addressed by deleting the intermediate nodes $(v\_1,\cdots,v\_\{n-1\})$ or $(v'\_1,\cdots,v'\_\{n-1\})$. However, deleting these nodes decreases the probability of exploring from these nodes, leading to inefficient exploration.
>
>   2.  We have conducted additional experiments to demonstrate that the proposed state abstraction function does not affect the convergence and performance of the algorithm when using larger simulation counts. Please refer to Figure 3 in the uploaded PDF for the experimental results. One possible reason is that more simulations lead to more accurate estimated node values, which makes the judgment of state abstraction functions more accurate.
>
> **Questions:**
>
> 1. We apologize for the lack of detail in our paper regarding the pruning operation. The $pruning(b_j)$ action denotes removing unique nodes of path $b_j$ compared to the other abstracted path in the search tree. We have provided more detailed explanations of the add/delete operations used in Algorithm 1. The specific modifications can be found in the general response.
>
> 2. Thanks for your suggestion. The requirement of the same best action is a very interesting and useful property, and it is consistent with state abstraction $\phi^\{\epsilon\}_\{a^*\}$ described in Table 1. This property may help to improve the robustness of the existing state abstraction functions. We plan to explore and combine this property with our proposed state abstraction method in future work. Furthermore, we will cite and discuss this work [1] in the final version.
>
> [1] Are AlphaZero-like Agents Robust to Adversarial Perturbations?

---

> > ### Comment · Reviewer_GJR1 · 2023-08-20
> > **Official comment**
> >
> > Weaknesses:
> >
> > 1.1 Even when utilizing PTSA, there's potential for incorrect aggregations. Consider sequences  (s1, s2, s3, ..., s_n), (s1, s2, s3, ..., s_n-1, s'_n), if v(s_n) is similar with v(s'_n), then PTSA may aggregate them. However, v(s'_n) might be wrong.
> >
> > 1.2 For MCTS, it is acceptable to have two paths leading to the same node (s1->s2->s3, s1->s4->s3) as long as there is no cycle (s1->s2->s3->s1). This approach enables the aggregation of more nodes.
> >
> > 2 You need to compare baselines with PTSA using the same simulation. For example, PTSAZero n=30 looks worse than MuZero n=30 in Pong.
> >
> > Questions:
> >
> > 2 Sorry for not making [1] clear enough.  Their main concept is that V can be wrong but will improve after looking forward (evaluating the next state).
> >  Let $T(s, a)$ be the transition function. [1] will require $V(T(s_1,a^*_1 )), V(T(s_1,a^*_2 )) , V(T(s_2,a^*_1 )) , V(T(s_2,a^*_2 ))$ to have a similar value. It can be extended from the optimal action to the optimal path.

---

> > > ### Author Response · Authors · 2023-08-21
> > >
> > > We appreciate your valuable comments and will address these points in the following response.
> > >
> > > **Weakness**
> > >
> > > 1.1  Completely eliminating errors in aggregation remains a challenge, and existing state abstraction methods cannot guarantee the absence of incorrect aggregations. We have emphasized that our method aims to improve the fault tolerance of abstraction during the training process, rather than guaranteeing the absence of incorrect aggregations, which is consistent with the results presented in Section 5.2 of our experiments.
> > >
> > > 1.2 In MuZero, the nodes within the search tree represent hidden states rather than explicit states, posing challenges in determining the appropriate nodes to aggregate. Following your recommendations, we compare the performance of PTSAZero with the modified version (consider two similar nodes not only paths) in Pong, Freeway, and Boxing tasks. Results are given in the following table:
> > >
> > > | Methods  |Task: Time-Return  | 60 mins | 120 mins | 180 mins | 240 mins | Average Reduction Rate |
> > > | -------- | ------- | ------- | ------- | ------- | -------- | -------------- |
> > > | PTSAZero N=18 | Pong |  8.7 |    16.8     |     17.5    |    18.1      |    16.5%            |
> > > |          | Freeway |   27.8      |    28.3     |     28.8    |     27.1     |        10.6%        |
> > > |          | Boxing|  35.4        |   62.5        |    89.1       |  90.2    |     41.3%          |
> > > | PTSAZero (modified)  N=18 | Pong    |    -8.6     |    -4.4     |     1.7    |    8.2      |     72.8%    |
> > > |          | Freeway |     14.2    |      18.8   |     20.5    |    26.1      |  69.2%    |
> > > |          | Boxing|     25.4    |    40.7     |   53.9        |      67.0                |    52.2%     |
> > >
> > > The results demonstrate that although there are more nodes to be aggregated, more incorrect aggregations may lead to worse performance.
> > >
> > > 2 Figure 1 in the uploaded PDF shows the comparison in Assault, MsPacman, Breakout, and Seaquest tasks with PTSA using the same simulation, which demonstrates that  PTSAZero N=30 achieves comparable performance as MuZero N=30 with less training time. The following Table presents the results in Pong, Freeway, and Boxing tasks:
> > >
> > > | Methods  |Task: Time-Return  | 60 mins | 120 mins | 180 mins | 240 mins |
> > > | -------- | ------- | ------- | ------- | ------- | -------- |
> > > | PTSAZero N=30 | Pong |  -11.2 |    4.5     |     18.5   |    19.2      |
> > > |          | Freeway |  16.7     |   27.3     |     28.5    |    27.0     |
> > > |          | Boxing|  37.2        |   64.9        |   68.5       |  81.3    |
> > > | MuZero N=30 | Pong    |    -3.5     |    11.3     |    16.3     |    18.0      |
> > > |          | Freeway |     19.0    |    24.5     |     27.3    |    26.5      |
> > > |          | Boxing|     31.4    |    58.5     |   64.0        |      70.6         |
> > >
> > > **Question**
> > >
> > > 2  Thanks for providing additional clarification for [1]. It appears that their main concept revolves around the idea that the value function V may initially be incorrect but can improve through forward-looking evaluations of the next state. PTSA compares the nodes at different depths along the path, which shares some similarities with the method described in [1]. Our contributions focus on extending the theory of state abstraction to tree structures and ensuring the transitivity of state abstractions within the tree space. Our method takes a different perspective compared to the method described in [1]. We will discuss [1] in the final version.

---

> > > > ### Comment · Reviewer_GJR1 · 2023-08-22
> > > > **Official comment**
> > > >
> > > > Thank you for the new table. According to the new table, Muzero is better for the first 120 mins in Pong and is better for the first 60 mins fin Freeway. Can you discuss more about them? Thank you.
> > > >
> > > > For weakness 1.1, I wonder if you have thought about undoing the aggregation after a while of searching. For example, you merge s, s' since V(s) and V(s') are similar, but after a while v(s) is different, maybe you can undo the aggregation.

---

> > > > > ### Author Response · Authors · 2023-08-22
> > > > >
> > > > > Thank you for your response. We are glad to discuss your concerns.
> > > > >
> > > > > Regarding the new table, MuZero has shown good learning efficiency in the early stages of training in simple environments, such as Pong and Freeway. However, PTSAZero may not improve training efficiency during the early stages of training due to less accurate estimations from the value and reward networks. As the network converges, PTSAZero's training efficiency will increase, allowing it to reach the convergence faster than MuZero. Expeimental reults show that in more complex environments such as Seaquest and MacPacman, PTSAZero can achieve higher efficiency even in the early stages of training.
> > > > >
> > > > > A similar undoing aggregation operation has been considered in our method. As the value estimation becomes more accurate, some nodes that were previously aggregated in the search will no longer be aggregated in the new search. This is because with each new timestep, the value of the search tree nodes is re-evaluated, leading to changes in the aggregation results.

---

### Official Review · Reviewer_CrLf · 2023-07-26

**Soundness:** 3 good
**Presentation:** 1 poor
**Contribution:** 3 good
**Rating:** 7
**Confidence:** 3

**Summary:**

This paper suggests a method of abstracting the state space explored by a Monte Carlo Tree Search (MCTS) algorithm, in order to reduce the complexity of the search. We can create an abstraction of the state space for MCTS by considering an abstraction over entire paths in the tree - two paths of equal length, that start from the same node in the tree, can be aggregated into a single abstract state, thus reducing the search space. The paper proposes to use a probabilistic approach to the abstraction process, using the justification that this enables the algorithm to recover from aggregation errors that it commits early on. The specific probabilistic approach discussed relies on a divergence measure between the distribution of the value functions across the entire two paths, thus merging together with high probability actions that lead to similar distributions of the value function. This abstraction helps mitigate the worst weakness of MCTS - it reduces the large search space. Some theoretical guarantees are provided, as well as several experimental results for different game problems and for control tasks.

**Strengths:**

The paper deals with the very important challenge of improving MCTS techniques. This type of research is applicable in many domains, as this is a well known and well used algorithm.

The experimental results looks extensive and well-presented, and are the main strength of the paper. I especially liked the comparison of different state abstraction functions, as it showcases the contribution of the paper in coming up with a specific one that seems to work well. Adding a successful novel approach to a well established algorithm is not a trivial task, and experimental results seem very promising. This seems like a strong enough reason to publish on its own.

**Weaknesses:**

I thought the main weakness of the paper is its readability. I had a tough time understanding the approach and the logic behind it, even though I have some experience with MCTS specifically (though admittedly, it had been awhile). Some more careful attention can be given to notation and explanations. The math in this paper requires close scrutiny and some of the explanations seem to assume a close familiarity with the specifics of MCTS, as well as state abstraction functions. This results in a reduced reading experience and lower comprehension.
Some examples:
1. In equation (1) Q is never explicitly defined, figure 1 appears much earlier in the paper than the definition of the probability state abstraction
2. The complex distinction between paths, states and nodes is not explicitly stated, and sometimes ignored - table 1 is referenced during the general RL discussion that has a state notation (s1, s2) but uses a different notation, that is later used for nodes (v1, v2).
3. Some of the notation within Algorithm 1 is never explained (e.g., actions like pruning/delete/add and the usage of a hidden state h).
4. Q* usage is never explained
5. In the explanation after definition 4.3 - encourages nodes that have the same candidate actions with similar value distribution expectations to be aggregated - should that be be encourages paths ? The entire definition seems to be about aggregating paths instead of specific states, but paths that start from the same node.

It is fine to delegate some details to referenced works, but a paper should at least succinctly explain its own notations and be as self-contained as possible. I trust these weaknesses in explanations and paper organization can be fixed by the authors.

**Questions:**

1. Are you planning to publish the code you used?
2. Please check your math for some typos - eq. 19 in appendix A.


**Limitations:**

Some limitations are briefly addressed, namely the need for hyper-parameter tuning and manually selecting the underlying abstraction function. I believe another limitation may lie in the added computational complexity of this method.

---

> ### Author Rebuttal · Authors · 2023-08-07
>
> Thank you for your uplifting review and valuable feedback. We appreciate your comments and would like to address them below.
>
> **Weaknesses:**
>
> Thanks for your feedback on the paper's readability. We will include more clear and coherent explanations of our method to improve the readability.
>
> 1. $Q(s, a)$ denotes the value of action $a$ in state $s$. We will explicitly define all variables and concepts used in the final version. Moreover, we will revise the paper to ensure that the definition of the probability state abstraction is presented before Figure 1.
> 2. We will provide more clear and explicit definitions for the concepts of paths, states and nodes in the final version: A path is a sequence of nodes in the search tree; A node in the search tree denotes the representation of the corresponding state. According to your advice, we will use the more general state notation $s$ as the input to the abstraction function in Table 1.
> 3. We apologize for not providing a clear explanation of the pruning/delete/add actions in the paper. We have provided more detailed explanations of the actions and notation used in Algorithm 1:
> $S_L $ is a list that records the searched paths in the current search tree. $S_L.delete(b)$ and $S_L.add(b)$ refer to removing and recording path $b$ in $S_L$ respectively. The $pruning(b_j)$ action denotes removing unique nodes of path $b_j$ compared to the other abstracted path in the search tree.  $h$ denotes the hidden feature of the original real environment state, which is employed to prevent MCTS from interacting with the actual environment during simulation in MuZero.
>
>
> 4. $Q*(s,a)$ denotes the value of optimal action in state $s$.
> 5.  We appreciate and agree with your opinion that using "path" instead of "state" is more appropriate. We will make the necessary changes in the final version.
>
> **Questions:**
>
> 1. We will release our code upon acceptance, and also conduct further research to explore how different state abstraction functions can be better tailored to fit the characteristics of the environment. We hope that this research will help to enhance the performance and applicability of our method in a wider range of applications.
> 2. Thanks for your careful review. We have corrected the typos in Appendix Equation (19):
>   \begin{equation}
>   p_{bM}\left(b_{1}, b_{2}\right) \wedge p_{bM}\left(b_{2}, b_{3}\right)
>   = p_{vM}\left(v_{1}, v_{3}\right) \wedge p_{vM}\left(v_{2}, v_{4}\right) \wedge p_{vM}\left(v_{3}, v_{5}\right) \wedge p_{vM}\left(v_{4}, v_{6}\right)
>   = p_{vM}\left(v_{1}, v_{5}\right) \wedge p_{vM}\left(v_{2}, v_{6}\right) \\
>   = p_{bM}\left(b_{1}, b_{3}\right).
>   \end{equation}
>   We really appreciate your feedback.
>
> **Limitations:**
>
> Thanks for your feedback on the limitations of our work. To further discuss the limitation of the added computational complexity, we have conducted more experiments including 9 Atari games to further discuss the limitation of the added computational complexity. Experimental results (shown in Table 1 of the uploaded PDF) demonstrate that PTSA introduces an acceptable decrease in trajectory collection efficiency (less than 8% on average), which results in a significant reduction in the whole training time.

---

### Author Rebuttal · Authors · 2023-08-07

## General Response ##

We thank all reviewers for their valuable feedback. We have carefully considered your suggestions and conducted additional experiments (shown in the uploaded PDF) to address your concerns, as outlined below:

1. We have conducted more Atari experiments (Assault, Seaquest, Breakout, and MsPacman tasks). The experimental results can be found in Figure 1 (in the uploaded PDF), which demonstrate the effectiveness of our algorithm in training acceleration.
2. We have conducted more experiments to evaluate the aggregation percentage in Atari, Gomoku and Control Tasks. The experimental results can be found in Figure 2 (in the uploaded PDF), which further validate the reduction of the tree search space resulting from our proposed algorithm.
3. To compare the effectiveness and aggregation percentage of PTSAZero with different numbers of simulations ($N$), we have
 conducted additional experiments as shown in Figure 2&3 (in the uploaded PDF). We observe that increasing the number of simulations does not significantly affect the aggregation percentage for the same environment and state abstraction function.
4. As shown in Table 1 (in the uploaded PDF), we have also conducted an analysis of the time consumption. PTSA introduces an acceptable decrease in trajectory collection efficiency (less than 8% on average), which results in a significant reduction in the whole training time.


Furthermore, we have improved the presentation to make our paper more comprehensible, which will be incorporated into the final version:

(1) We provide more detailed explanations of the notations used in Algorithm 1, including the pruning/delete/add actions and the hidden state $h$:

$S_L$ is a list that records the searched paths in the current search tree. $S_L.delete(b)$ and $S_L.add(b)$ refer to removing and recording path $b$ in $S_L$ respectively. The $pruning(b_j)$ action denotes removing unique nodes of path $b_j$ compared to the other abstracted path in the search tree.  $h$ denotes the hidden feature of the original real environment state, which is employed to prevent MCTS from interacting with the actual environment during simulation in MuZero.

(2) We have corrected a typo in Appendix Equation (19). The corrected version is as follows:
\begin{equation}
    p_{bM}\left(b_{1}, b_{2}\right) \wedge p_{bM}\left(b_{2}, b_{3}\right)
    = p_{vM}\left(v_{1}, v_{3}\right) \wedge p_{vM}\left(v_{2}, v_{4}\right) \wedge p_{vM}\left(v_{3}, v_{5}\right) \wedge p_{vM}\left(v_{4}, v_{6}\right)
    = p_{vM}\left(v_{1}, v_{5}\right) \wedge p_{vM}\left(v_{2}, v_{6}\right)
    = p_{bM}\left(b_{1}, b_{3}\right)
\end{equation}

Finally, we would like to express our sincere gratitude to all the reviewers for their valuable suggestions. The code will be released upon acceptance. We hope that our response will resolve your concerns.

---

### Decision · Program_Chairs · 2023-09-21

**Decision:**

Accept (poster)

**Comment:**

The reviewers found the paper to make a solid contribution. Please take the critical feedback into account when preparing the camera ready version.